# A modular CRISPR screen identifies individual and combination pathways contributing to HIV-1 latency

Emily Hsieh[1], Derek H. Janssens[2], Patrick J. Paddison[3], Edward P. Browne[4], Steve Henikoff[2,5], Molly OhAinle[3¤], Michael Emerman[2,3]*

1 Molecular and Cellular Biology Graduate Program, University of Washington, Seattle, Washington, United States of America, 2 Division of Basic Sciences, Fred Hutchinson Cancer Center, Seattle, Washington, United States of America, 3 Division of Human Biology, Fred Hutchinson Cancer Center, Seattle, Washington, United States of America, 4 Division of Infectious Diseases, Department of Medicine, University of North Carolina at Chapel Hill, Chapel Hill, North Carolina, United States of America, 5 Howard Hughes Medical Institute, Chevy Chase, Maryland, United States of America

¤ Current Address: Division of Immunology and Molecular Medicine, Department of Molecular and Cell Biology, University of California, Berkeley, California, United States of America
* memerman@fredhutch.org

**Data Availability Statement:** Raw data for CRISPR screens are available at https://www.ncbi.nlm.nih.gov/geo/query/acc.cgi?acc=GSE213995 Raw sequencing data for the CUT&Tag experiments are

## Abstract

Transcriptional silencing of latent HIV-1 proviruses entails complex and overlapping mechanisms that pose a major barrier to *in vivo* elimination of HIV-1. We developed a new latency CRISPR screening strategy, called Latency HIV-CRISPR which uses the packaging of guideRNA-encoding lentiviral vector genomes into the supernatant of budding virions as a direct readout of factors involved in the maintenance of HIV-1 latency. We developed a custom guideRNA library targeting epigenetic regulatory genes and paired the screen with and without a latency reversal agent–AZD5582, an activator of the non-canonical NFκB pathway–to examine a combination of mechanisms controlling HIV-1 latency. A component of the Nucleosome Acetyltransferase of H4 histone acetylation (NuA4 HAT) complex, ING3, acts in concert with AZD5582 to activate proviruses in J-Lat cell lines and in a primary CD4+ T cell model of HIV-1 latency. We found that the knockout of *ING3* reduces acetylation of the H4 histone tail and BRD4 occupancy on the HIV-1 LTR. However, the combination of *ING3* knockout accompanied with the activation of the non-canonical NFκB pathway via AZD5582 resulted in a dramatic increase in initiation and elongation of RNA Polymerase II on the HIV-1 provirus in a manner that is nearly unique among all cellular promoters.

## Author summary

HIV-1 establishes long-lived latent reservoirs that present a barrier for virus eradication and HIV Cure. One approach to reduce the latent reservoir is the use of small molecules called latency reversal agents (LRA) to trigger immune-mediated clearance of virus-producing cells. However, this approach is limited by the inability of current LRAs to reactivate the majority of latent proviruses and a lack of specificity, highlighting the need for a

available at https://www.ncbi.nlm.nih.gov/geo/query/acc.cgi?acc=GSE215430 All other data is available in the supplemental tables of the manuscript. Raw data for all experiments in S8 Table. CRISPR screen data and primer information in S1, S2, S3, S5, S6 and S7 Tables

**Funding:** This work was funded by grants from the NIH: DP1 DA051110 (ME), UM1-AI164567 (ME, EPB), T32 AI 083203 (EH), DK 119270 (PJP), DK106829 (PJP), AI143381 (EPB), DA047023 (EPB), and AI147877 (MO); NSF: DGE-1256082 (EH) and DGE-176211 (EH); Hartwell Foundation Postdoctoral Fellowship (DHJ); and the Howard Hughes Medical Institute (SH). The funders had no role in study design, data collection and analysis, decision to publish, or preparation of the manuscript.

**Competing interests:** The authors have declared that no competing interests exist.

greater understanding of the interplay of mechanisms involved in the maintenance of HIV-1 latency. We developed a modular CRISPR-based screening, called Latency HIV-CRISPR, in a T-lymphocyte (J-Lat) model of latent infection to gain a comprehensive representation of the pathways involved in HIV-1 latency. As the latent state of HIV-1 is controlled by multiple, parallel mechanisms, we used the screen to study the epigenetic factors in the presence of an LRA targeting a transcriptional activation mechanism. The screens identified novel factors controlling HIV-1 latency both in the J-Lat and primary T cell models of HIV-1 latency.

## Introduction

Effective antiretroviral therapy (ART) can drive HIV-1 viral loads to undetectable levels [1]. However, ART does not eliminate the virus from people living with HIV-1. Upon interruption of ART, there is a rapid rebound of virus replication from a long-lived latent reservoir primarily found in memory CD4+ T cells among other cell types [2]. This latent reservoir of replication-competent HIV-1 proviruses is a significant obstacle to the complete clearance of HIV-1 [3], and purging or managing this latent reservoir is essential to achieve a functional cure.

The latent HIV-1 reservoir is often transcriptionally silent but can produce infectious virus upon T-cell activation [4]. The transcriptional silencing of the provirus in HIV-1 latency is dependent on integration site [5, 6] and a breadth of host factors that limit transcriptional initiation and elongation of the HIV-1 proviral genome (reviewed in [7]). Small molecule inhibitors or activators, known as latency reversal agents (LRAs), have been shown to activate HIV-1 viral transcription in cell line models of HIV-1 latency [8] and in primary cell cultures derived from people living with HIV-1 [9, 10]. AZD5582 is a second mitochondria-derived activator of caspases (SMAC) mimetic that is an example of an LRA that provides some specificity for latency reversal by activating the non-canonical NFκB pathway [11]. Many LRAs have shown success at reactivating latent HIV-1 in HIV-1 latency models, but even the most well-established LRAs have modest or no effect in clinical trials on their own [12]. This highlights the complexity of mechanisms involved in maintaining the HIV-1 latent state, and the need to identify novel epigenetic factors and transcriptional mechanisms controlling HIV-1 latency.

Studies on the functional role of host epigenetic regulation of HIV-1 transcription (reviewed in [7, 13, 14]) has resulted in the identification of some specific host epigenetic regulatory genes that are involved in maintaining HIV-1 latency. For example, KAT5 of the Nucleosome Acetyltransferase of H4 histone acetylation (NuA4 HAT) complex deposits a uniquely high profile of acetylated lysine residues on the H4 histone tail (H4Ac) found at the HIV-1 provirus LTR [9]. This results in the recruitment of the long isoform of the *Bromodomain Containing 4* (*BRD4*) gene [9, 15, 16]. Additionally, the long BRD4 isoform in *in vitro* experiments serves as a competitor for positive transcription elongation factor b (P-TEFb) binding to the HIV-1 transactivator, Tat [16]. *BRD4* also encodes a short isoform that recruits the repressive BAF complex to the HIV-1 provirus [17]. Another set of genes implicated in HIV-1 latency through repression of HIV-1 transcription, encode components of the Polycomb Repressive Complex 2 (PRC2) [10, 18, 19]. PRC2 catalyzes the methylation of histone H3 Lysine 27 to maintain transcriptional repression of genes, including throughout development. Additionally, epigenetic regulatory silencing mechanisms also target unintegrated HIV-1 DNAs, such as CHAF1A and CHAF1B [20].

The use of genetic screens has been effective in revealing pathways involved in HIV-1 latency in a single-gene manner [10, 21–28]. Here, we sought to create a novel CRISPR-based

screening platform with the capacity to allow for rapid, parallel assessment of multiple pathways that contribute to the maintenance of HIV-1 latency both with and without the presence of additional LRAs. We reasoned that by combining LRAs of one mechanism with a CRISPR screen that targets a different mechanism, we could uncover latency reversal pathways with increased potency and specificity for the HIV-1 LTR. We established a novel HIV-based latency CRISPR screening strategy that uses J-Lat cells [29, 30] in combination with our recently developed HIV-CRISPR screening methodology [31] to use reactivation of the latent provirus as the reporter for the screen. To identify epigenetic pathways that act with and without LRAs to activate latent proviruses, we generated a custom CRISPR library targeting epigenetic regulatory genes and performed the screen in the absence and presence of a low-activating dose of AZD5582. In the absence of AZD5582, we found *CUL3*, a scaffold for an E3 ubiquitin ligase, as a novel HIV-1 latency maintenance and establishment factor in both J-Lat cells and a primary CD4+ T cell model of HIV-1 latency. Moreover, we identified the NuA4 HAT complex to be of importance to HIV-1 latency. In particular, knockout of *Inhibitor of Growth Family Member 3 (ING3)*, a subunit of the NuA4 complex, combined with AZD5582 treatment resulted in enhanced viral reactivation in the J-Lat cells and a primary CD4+ T cell model of HIV-1 latency. By using automated CUT&Tag to investigate genome-wide chromatin occupancy [32], we found that *ING3* knockout alone and in combination with AZD5582 treatment results in the reduction of H4Ac and BRD4 levels at the HIV-1 LTR. However, only the addition of AZD5582 in the context of the *ING3* knockout resulted in a substantial increase in RNA Polymerase II Serine 5 phosphorylation (RNA-Pol2-S5p) occupancy at the LTR in a manner that was nearly unique among all promoters in the human genome. Simultaneously, the combination of *ING3* knockout and AZD5582 treatment results in increases in RNA-Pol2-S5p and RNA-Pol2-S2p within the body of the provirus, which indicates increased transcription initiation and elongation and correlates with the increase of viral reactivation. Our novel Latency HIV-CRISPR screening approach provides an avenue to explore factors that act in combination to promote HIV-1 latency.

## Results

### Latency HIV-CRISPR identifies novel epigenetic factors involved in maintaining HIV-1 latency

Our aim was to establish a high throughput CRISPR-Cas9 knockout strategy for investigating the complexities of epigenetic regulation of HIV-1 latency. We leveraged a previously described HIV-CRISPR vector [31] which is a lentiviral based vector with two intact LTRs, a packaging signal, Cas9, and a library of single guide RNAs (sgRNAs). The unique feature of the HIV-CRISPR vector is that in the presence of replication-competent HIV-1, the genome of the HIV-CRISPR vector is transcribed into RNA and packaged *in trans* into HIV-1 virions. We reasoned that in J-Lat cells, an *in vitro* T-lymphocyte HIV-1 latency model which harbors a near full-length HIV-1 provirus, any sgRNAs encoded in HIV-CRISPR genomes that target candidate HIV-1 latency genes will be packaged into the reactivated virus particle, secreted, and enriched in the viral supernatant (**Fig 1A**). Consequently, the RNA of the viral supernatant from the HIV-CRISPR latency screen can undergo next-generation sequencing to determine sgRNAs that are enriched in the viral particles and thereby identify candidate genes involved in HIV-1 latency maintenance. This results in a direct readout for the gene knockout which incorporates functional, reactivation activity.

To specifically investigate host epigenetic regulators involved in the maintenance of HIV-1 latency, we generated a custom human epigenome specific sgRNA CRISPR library (HuEpi). This library contains sgRNAs targeting epigenome factors such as histones, histone binders

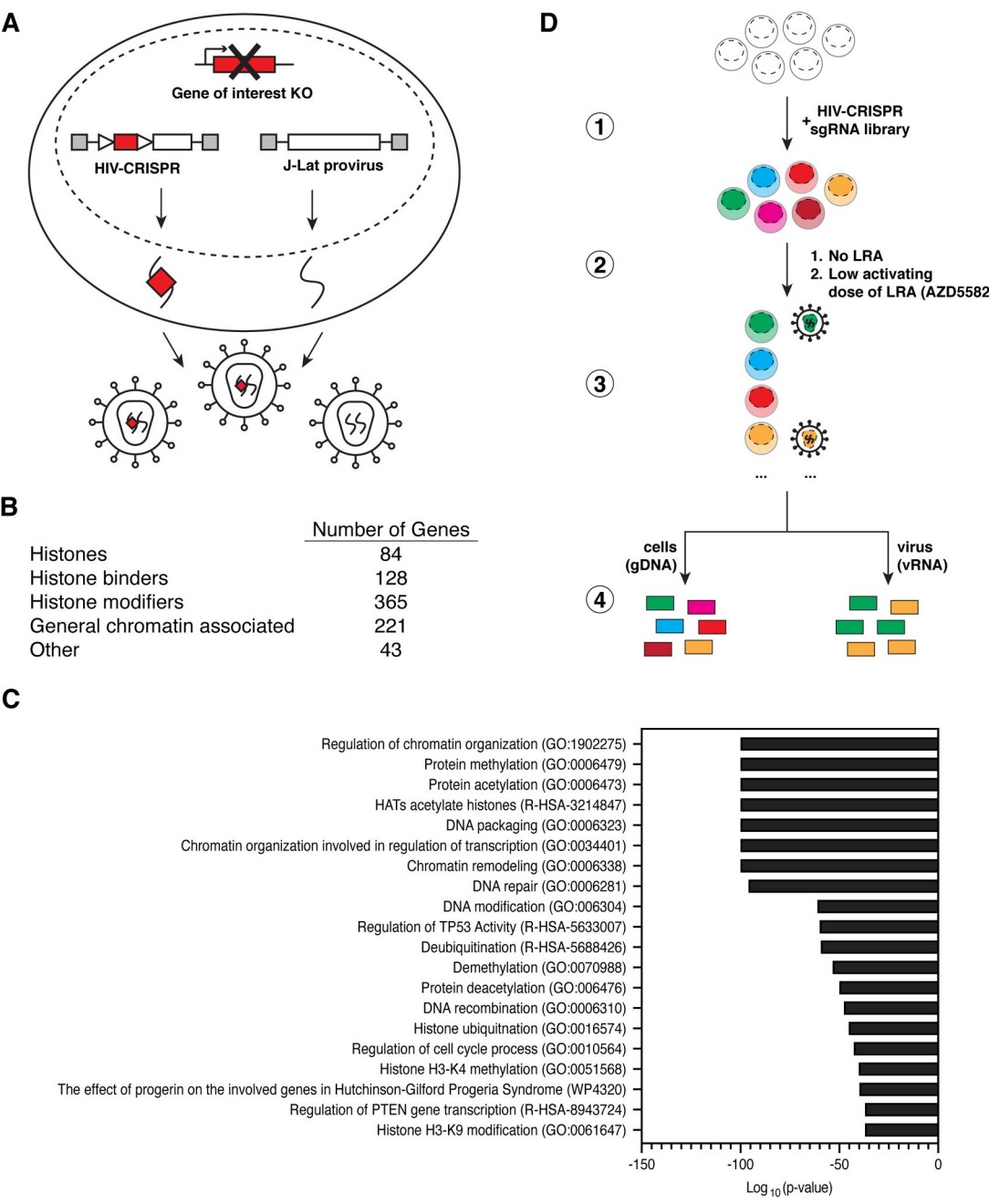

**Fig 1. The Latency HIV-CRISPR screen.** (**A**) Schematic summarizing the Latency HIV-CRISPR screen in J-Lat cell lines showing the latent, integrated provirus and the HIV-CRISPR vector which delivers Cas9 and a sgRNA and produces a packageable genomic RNA. Red boxes represent the gene or sgRNA targeting the gene of interest; gray boxes represent functional HIV-1 LTRs at both sides of the vector and provirus; triangles represent internal promoters for sgRNA and Cas9 transcription. (**B**) Categorical distribution of the genes targeted by the Human Epigenome (HuEpi) sgRNA library. (**C**) Metascape Gene Ontology (GO) analysis [44] of the genes targeted by the HuEpi sgRNA library. (**D**) Summary of workflow from the generation of the pool of J-Lat HuEpi knockout cells to the sequencing and comparison of the abundance of guideRNAs found in the viral RNA (vRNA) pool versus the genomic DNA (gDNA) pool. GuideRNAs enriched in the viral supernatant RNA relative to the genomic DNA represent the gene(s) that upon knockout result in latency reversal.

(e.g., histone readers and chaperones), histone modifiers (e.g., histone writers and erasers), and general chromatin associated factors (e.g., RNA and DNA modifiers) (**Fig 1B and 1C**).

Most of the genes targeted in this library were derived from the EpiFactor database [33]. To this set we also added histones and other hand-selected gene regulatory complexes. The total library contains 5,309 sgRNAs targeting 841 genes (6 sgRNAs per gene with a few exceptions) and 252 non-targeting controls (NTCs) [34] (**S1 Table**). The small size of the library improves the likelihood of high coverage of all the sgRNAs throughout the CRISPR screening process relative to a whole-genome screen.

The sgRNA sequences from the HuEpi library were cloned in bulk into the HIV-CRISPR vector and the lentiviral library was subsequently transduced into J-Lat cells at a low MOI (MOI = 0.4) (**Fig 1D**-**1**). We transduced two independent J-Lat cell lines, the J-Lat 10.6 [29] and J-Lat 5A8 [30] cells, which each have different provirus integration sites. The J-Lat 5A8 cells respond to LRAs more similarly to primary CD4+ T cell HIV-1 latency models compared to other cell line models [35]. Furthermore, as knockout of the host restriction factor *Zinc Antiviral Protein* (*ZAP*) has been demonstrated to improve the performance of the screen as a result of ZAP-mediated inhibition of the HIV-CRISPR vector [31], we generated and performed the screen in clonal *ZAP* knockout J-Lat 10.6 and J-Lat 5A8 cells. Transduced cells were then subjected to 10–14 days of puromycin selection after which the cells and viral supernatant were harvested and processed by next-generation sequencing of the sgRNAs to identify the candidate HIV-1 latency genes (**Fig 1D**-**3**). Genomic DNA was extracted from the cell pellet to determine the baseline representation of each sgRNA in the transduced sgRNA library. Viral RNA was extracted from the concentrated viral supernatant to identify the sgRNA population that facilitated latent HIV-1 provirus transcription reactivation (**Fig 1D**-**4**). Using the MAGeCK pipeline [36], we identified candidate HIV-1 latency maintenance factors by selecting genes in which the sgRNA counts of the viral RNA were enriched compared to the sgRNA counts in the genomic DNA. Genes with an increased sgRNA count in the viral RNA, which equates to an increase in the fold change, are predicted to play a role in maintaining HIV-1 latency.

Overall, the Latency HIV-CRISPR screen results for both J-Lat 10.6 and J-Lat 5A8 cell lines demonstrated a significant enrichment in specific sgRNAs compared to the NTCs. We compared the distribution of the log fold change (LFC) of NTCs against the gene targeting sgRNAs. While the LFC of most gene targeting sgRNAs clustered around the median LFC of the NTCs, we observed that a subset of the gene targeting sgRNAs were significantly different compared to the mean of the NTC population based on a Welch's t-test (**Fig 2A**). Our focus was the most highly enriched outlier sgRNAs as these are the sgRNAs of interest that target potential HIV-1 latency maintenance factor genes.

To determine top candidate genes from the Latency HIV-CRISPR screen, we ranked the genes by MAGeCK gene score. For both J-Lat cell lines, we observed that many genes were significantly enriched compared to the NTCs. We identified eight epigenetic HIV-1 latency maintenance genes as our top hits as defined by a false discovery rate (FDR) of less than 10% (FDR < 10%) in both J-Lat 10.6 and J-Lat 5A8 cell lines (**Fig 3A**). BRD4 and KAT5 are two host factors previously reported to silence HIV-1 transcription ([16, 17, 37] and [9], respectively), and in our screen, the sgRNAs targeting *BRD4* and *KAT5* are significantly enriched compared to the non-targeting sgRNAs in both J-Lat cell lines (**Fig 2B** and **S2 Table**). In addition to individual genes, the Latency HIV-CRISPR screen also identified genes encoding subunits of protein complexes important to HIV-1 latency. PRC2 has also been previously reported to silence HIV-1 transcription [8, 10] and members of the complex including *EZH2*, the catalytic component, and *SUZ12* and *EED* showed a trend of enrichment in our screen (**Fig 2B** and **S2 Table**). Thus, our Latency HIV-CRISPR screen system identifies bona fide HIV-1 latency maintenance factors.

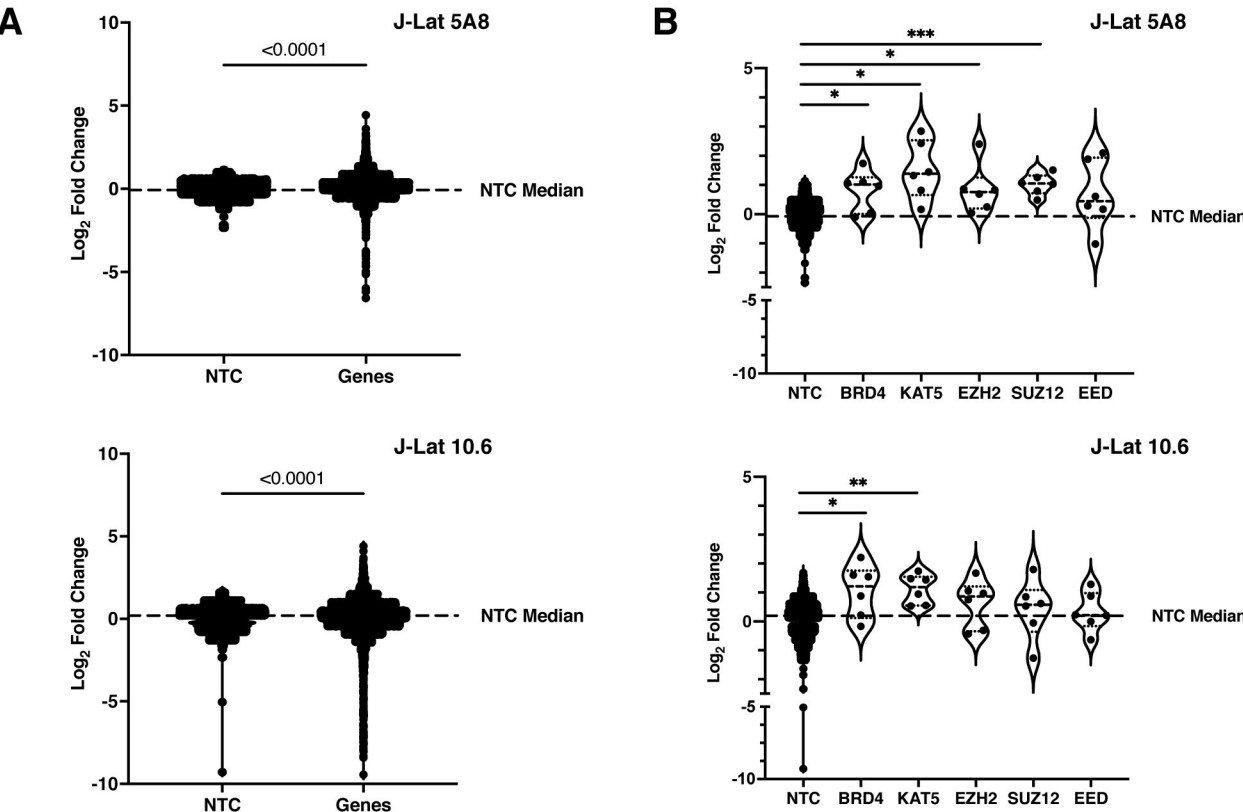

**Fig 2. GuideRNA level enrichment for known and novel genes of interest in the Latency HIV-CRISPR screen. (A)** GuideRNA enrichment ($\log_2$ of fold change) in both J-Lat cell lines for the genes targeted by the HuEpi sgRNA library compared to the Non-Targeting Control (NTC) guides. For statistical analysis, the HuEpi gene sgRNAs are compared to the NTC control. Welch's t-test, p-value < 0.0001. **(B)** GuideRNA level enrichment of known HIV-1 latency maintenance factors BRD4, KAT5, and PRC2 complex members (EZH2, SUZ12, and EED) in the Latency HIV-CRISPR screen. For statistical analysis, all conditions are compared to the NTC control. Each dot represents an individual guideRNA targeting the indicated gene. Welch's t-test, p-value = <0.05 = *, = <0.01 = **, = <0.001 = ***.

## The Latency HIV-CRISPR screen identifies multiple members of the NuA4 complex and CUL3

To identify the top candidate HIV-1 latency genes, we integrated the guideRNA data to a gene level using MAGeCK analysis [36] (**Fig 3A and S3 Table**). As we were primarily interested in hits that are independent of integration site, we considered the top gene hits as those that have an FDR < 10% and are shared between both J-Lat 10.6 and J-Lat 5A8 cell lines (**Fig 3B**, middle of Venn diagram). We found that a subset of genes had been previously implicated in the context of HIV-1 latency, while other top hits had not, to our knowledge, previously been studied in the context of HIV-1 latency (**S4 Table**).

We observed that multiple members of the Nucleosome Acetyltransferase of H4 (NuA4) complex scored highly in the screen (**Fig 3A**). The NuA4 complex is highly conserved in eukaryotes [38] and acetylates histone H4 [39] and has been implicated in many genomic processes including DNA damage repair and transcription [40]. Many proteins that are part of the NuA4 complex also overlap with the SNF2-Related CBP Activator Protein (SRCAP) complex which exchanges the H2A of the canonical nucleosome with the H2A.Z variant [41–43]. A gene ontology (GO) enrichment analysis [44] of the top gene hits (FDR <10% for each J-Lat cell line) demonstrated that the SRCAP-associated chromatin remodeling complex was highly enriched in both J-Lat cell lines (**Fig 3C**) and was different from the overall GO distribution of

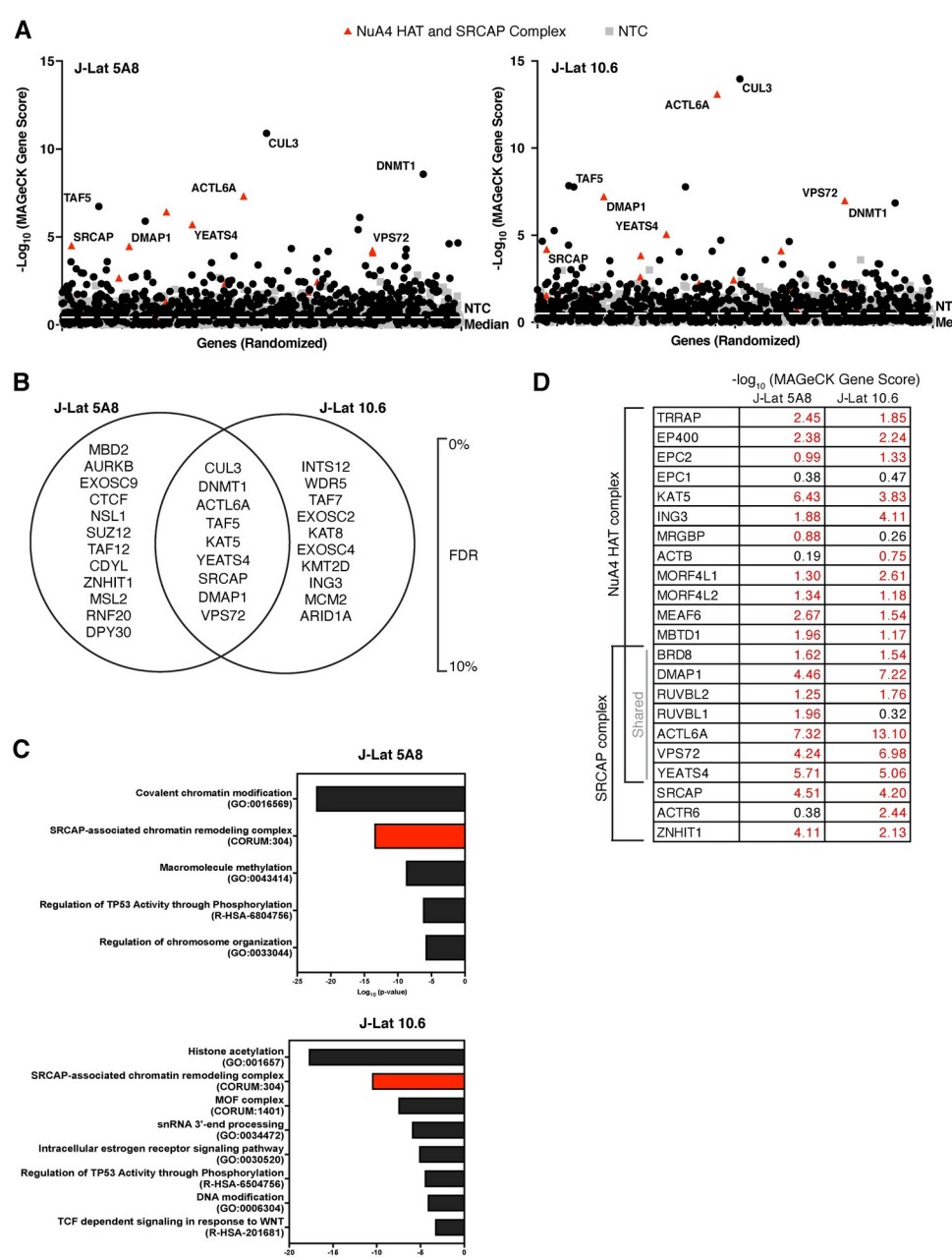

**Fig 3. The Latency HIV-CRISPR screen in J-Lat 10.6 and J-Lat 5A8 cells identifies a set of mutual, novel hits. (A)** The -log$_{10}$ of the MAGeCK scores for each gene targeted by the HuEpi sgRNA library (black circles and red triangles) and NTCs (gray squares) are calculated and displayed. Gene names are labelled for hits that have a <10% false discovery rate (FDR) in both J-Lat cell lines (center of Venn diagram in (B)); red triangles represent members of the NuA4 HAT complex and SRCAP complex. NTCs are artificial NTC genes designed by iterative binning of NTC sgRNA sequences (see Methods). Genes are randomized on the x-axis, but the same order is used for both right and left panels. The y-axis is the inverse log$_{10}$ of the MAGeCK score. **(B)** Top hit genes (<10% FDR) in common and unique to each J-Lat cell line are ordered by significance and FDR with the top of the list having the highest significance and lowest FDR. **(C)** Metascape GO analysis [44] of the gene hits with a <10% FDR in each J-Lat cell line. **(D)** Analysis of the -log$_{10}$ of the MAGeCK scores of the genes overlapping and unique to the NuA4 HAT and SRCAP complex compared to the NTCs. The higher the number, the more statistically significant it is of a hit. Red font is for genes that score higher than the average NTC score.

the total HuEpi library (**Fig 1C**), demonstrating the screen specifically enriched for specific pathways. Based on the MAGeCK gene scores, six of the seven genes (*BRD8*, *DMAP1*, *RUVBL2*, *ACTL6A*, *VPS72*, and *YEATS4*) that encode proteins overlapping between the NuA4 HAT and SRCAP complexes scored higher than the median MAGeCK score for the NTCs (**Fig 3D**). The only exception is *RUVBL1* in the J-Lat 10.6 screen, which scored below the NTC median MAGeCK score (RUVBL1 = 0.318 vs NTC median = 0.518). Additionally, most genes unique to each complex also had MAGeCK scores above the NTC median in both cell lines. Genes unique to the NuA4 HAT complex included *TRRAP*, *EP400*, *EPC2*, *KAT5*, *ING3*, *MORF4L1*, *MORF4L2*, *MEAF6*, and *MBTD1* and genes unique to the SRCAP complex included *SRCAP* and *ZNHIT1* (**Fig 3D**). This analysis suggests that the Latency HIV-CRISPR screen strategy can extend beyond identification of single genes of interest to groups of genes encoding complexes of relevance, which can further shed light on the mechanisms involved in maintaining HIV-1 latency.

To validate the top screen hits, with a bias towards those that had not previously been implicated in HIV-1 latency (**S4 Table**), we generated four individual J-Lat 10.6 and 5A8 knockout cell lines for each gene (two sgRNAs per gene). Because the transduced cells were assayed as a pool of gene knockouts, we simultaneously measured CRISPR editing efficiency in bulk using the program Inference of CRISPR Edits (ICE) [45]. The knockout efficiency for the heterogenous cell lines we generated varied between 32% to 83% (**Fig 4A**). A recent study demonstrated that a median knockout of 35% was less efficient, but still had a meaningful effect [45]. As positive controls, we used two sgRNAs targeting *NFκBIA*, an inhibitor for NFκB, and a gene that was not included in the HuEpi library. Upon knockout of *NFκBIA* in the J-Lat cells, we expected an increase in HIV-1 transcriptional activity, which leads to virus production. As negative controls, we used two NTC sgRNAs that do not target human loci. To assess latency reversal in the J-Lat cells, we measured HIV-1 reverse transcriptase activity in the supernatant, which is a readout of virus production after the viral transcription, translation, and budding steps have occurred. This quantification was performed 10–14 days after gene knockout and as the provirus in the J-Lat cells have a deletion in HIV-1 *env*, the reactivated virus particles do not spread in the culture. Also, the provirus of the J-Lat 5A8 cell lines was more challenging to reactivate compared to the provirus of the J-Lat 10.6 cell lines, which is consistent with the J-Lat 5A8 cell lines producing a reactivation profile more similar to primary CD4+ T cell models of HIV-1 latency [35].

Compared to the cell lines transduced with NTC sgRNAs, we found that individually knocking out all eight novel candidate gene hits resulted in significant latency reversal in at least one of the J-Lat cell lines we tested (**Fig 4A**). In fact, some of the reverse transcriptase measurements from the gene knockout cell lines are comparable to the positive control knockout of *NFκBIA*. In addition, we confirmed that knockout of a top scoring gene hit in our screen, *KAT5*, a previously reported HIV-1 latency maintenance factor [9], results in viral reactivation in both J-Lat cell lines (**Fig 4A**). Moreover, genes encoding protein components of the NuA4 HAT and SRCAP complexes described above, *ACTL6A*, *DMAP1*, *VPS72*, and *YEATS4* also all validated as latency maintenance factors in the J-Lat cells (**Fig 4A**). *ACTL6A* knockout results in a particularly high level of HIV-1 reactivation, which may be a result of *ACTL6A* encoding a protein that is a subunit of both NuA4 HAT and SRCAP complexes (**Fig 3D**). Other genes validated in this functional investigation include *DMAP1* which encodes a protein that forms a complex with DNMT1 to mediate transcriptional repression [46] and has been described to play a role in HIV-1 latency in some studies [14] and *TAF5* which functions in scaffold formation and is a critical subunit of the general transcription factor TFIID.

As *CUL3* was the top gene hit in our Latency HIV-CRISPR screen in both J-Lat 10.6 and J-Lat 5A8 cell lines and validated as a latency maintenance factor in J-Lat cells, we further

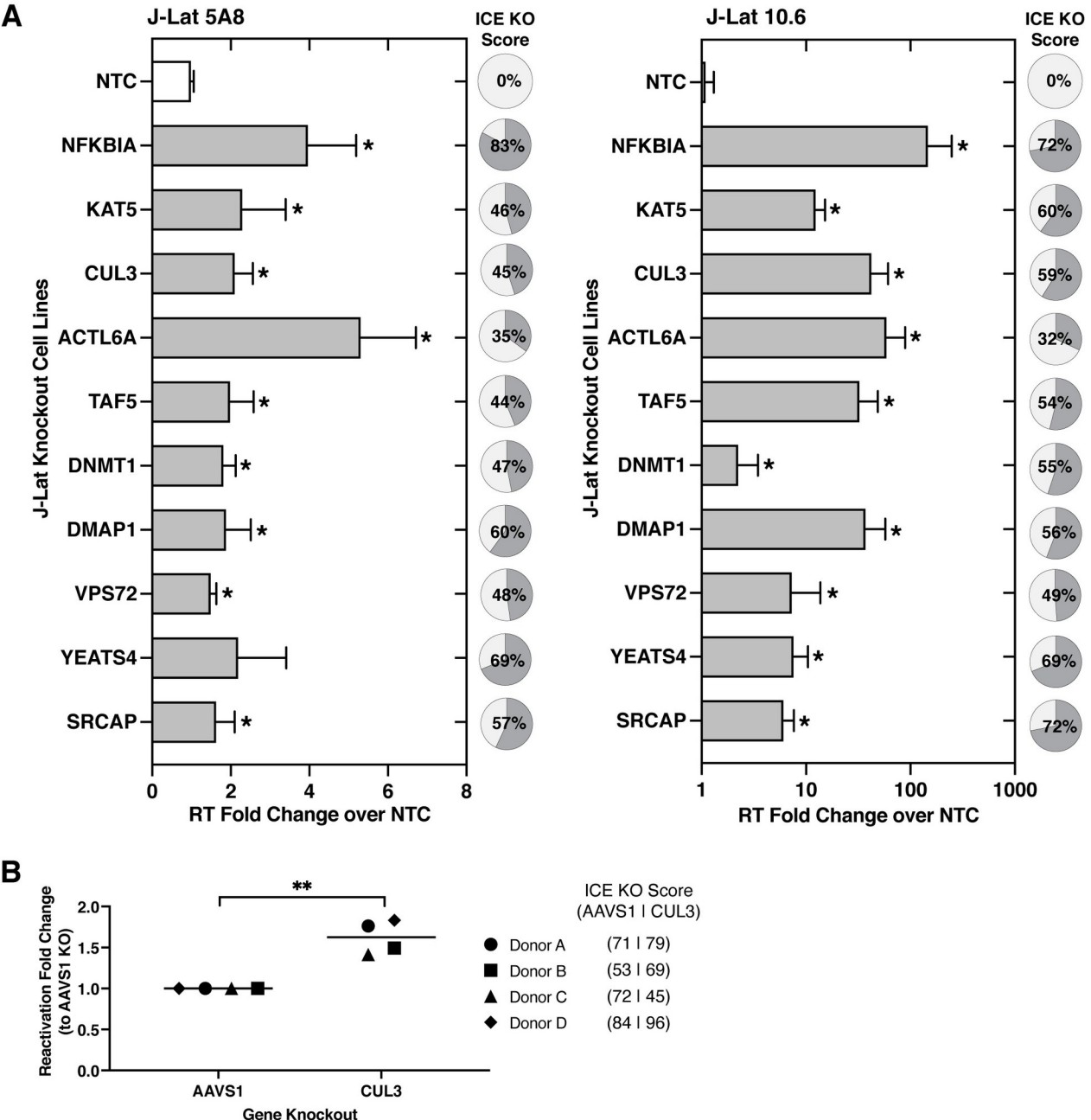

**Fig 4. Validation of top hits from Latency HIV-CRISPR screen. (A)** Validation of the top 9 gene hits of the Latency HIV-CRISPR screen was performed by individually knocking out each J-Lat cell line with two different guide RNAs and measuring viral reactivation by quantifying HIV-1 reverse transcriptase activity of the viral supernatant. *NFκBIA* knockout is a positive control. The reverse transcriptase activity from released virions was normalized to basal activity from the transduction of NTC sgRNA. For each for the knockout cell lines, ICE analysis was performed and the average knockout is shown in pie charts. Multiple unpaired t-tests, p-value = <0.05 = *. **(B)** *CUL3* knockout resulted in significant viral reactivation in a primary CD4+ T cell model of HIV-1 latency using cells from four healthy donors. Viral reactivation was measured by flow cytometry and normalized to the *AAVS1* knockout cells. Paired t-test, p-value = <0.05 = *, = <0.01 = **.

characterized *CUL3* in the context of a primary CD4+ T cell HIV-1 latency model system. CUL3 is part of the cullin family and serves as the scaffold for Cullin-RING E3 ligase complexes, which ubiquitinates proteins [47]. For our primary cell HIV-1 latency model system,

we used a dual HIV-1 reporter virus (pNL4-3-Δ6-dreGFP-CD90) that is derived from previously established systems [48, 49]. Here, the reporter virus has been modified to encode a destabilized eGFP gene and a mouse Thy1.2 (CD90.2) gene, which are separated by an IRES and both reporters are under control of the HIV-1 LTR [50]. As the mouse Thy1.2 reporter has a long half-life and is slow to turn over, it serves as a marker to determine the cells that have been infected by the reporter virus (Thy1.2+). Amongst the infected cells (Thy1.2+), the eGFP reporter, which has a short half-life, serves as a marker to determine the cells that are actively transcribing or actively infected (Thy1.2+, GFP+) versus the cells that are minimally transcribing or latently infected (Thy1.2+, GFP-). We knocked out *CUL3* in the primary CD4 + T cell HIV-1 latency model using Cas9-gRNA ribonucleoproteins (RNPs) and 3 different sgRNAs. As a negative control, the *AAVS1* gene was targeted for knockout, which is in a "safe harbor" locus in which disruption of this gene does not have adverse effects on the cell [51].

We expected that if CUL3 plays a role in HIV-1 latency establishment, there will be a higher reactivated population in the *CUL3* knockout cells compared to the negative control (*AAVS1*) knockout cells. We acquired CD4+ T cells from four healthy donors and infected the cells with the dual reporter virus. Indeed, upon performing the described CRISPR/Cas9-mediated knockout and measuring the functional output of the reporter virus, we observed that *CUL3* knockout in all four donors resulted in significant HIV-1 reactivation and subsequent decrease in the latent population compared to *AAVS1* knockout (**Figs 4B and S1A**). This suggests that post-translational ubiquitin modification of a substrate by CUL3 may facilitate the establishment and maintenance of HIV-1 latency. Moreover, these data demonstrate that the Latency HIV-CRISPR screening strategy is effective in identifying novel epigenetic HIV-1 latency maintenance genes that have been validated in multiple systems.

## An LRA Latency HIV-CRISPR screen reveals regulatory changes that act in combination during reactivation of HIV-1 latency

A critical extension of this Latency HIV-CRISPR screening methodology is the ability to examine other transcriptional mechanisms in conjunction with epigenetic regulatory factors. Thus, to address HIV-1 latency as a state of multiple, parallel mechanisms, we modified the screen by treating the pool of human epigenome (HuEpi) knockout cells with a low-activating dose of LRA. The goal was to identify instances of combinations between the gene knockout and LRA resulting in a significant increase in viral reactivation (i.e. a top hit in the screen). As a proof of principle, we used the recently identified LRA, AZD5582, which is a SMAC mimetic and non-canonical NFκB pathway activator [11]. We first determined 10 nM AZD5582 as the low-activating dose of LRA to use by performing an HIV-1 viral reactivation dose curve using both J-Lat cell lines (**S2A** Fig). The same HuEpi knockout pool of cells used in the Latency HIV-CRISPR screen described in **Fig 3** were in parallel treated for 24 hours with 10 nM of AZD5582 (**Fig 1D**-**2**). This allowed for the identification of epigenetic regulatory genes that upon knockout, combine with the transcriptional initiation pathway targeted by AZD5582, to result in a significant increase in viral reactivation.

The most prominent hit of the AZD5582 LRA Latency HIV-CRISPR screen in both J-Lat cell lines was *Inhibitor of Growth Family Member 3* (*ING3*), a novel HIV-1 latency factor (**Fig 5A and S5 and S6 Tables**). We compared the LRA Latency HIV-CRISPR screen results to the Latency HIV-CRISPR screen results in the absence of LRA. Overall, many gene hits overlapped between the Latency HIV-CRISPR screen in the presence and absence of AZD5582 (**Fig 5B**). For example, the previously identified *KAT5* gene and genes that we validated in **Fig 4** including *DNMT1*, *YEATS4*, and *ACTL6A*, all subunits of the NuA4 complex, were still amongst the top gene hits, which suggests that these genes broadly play a role in HIV-1 latency

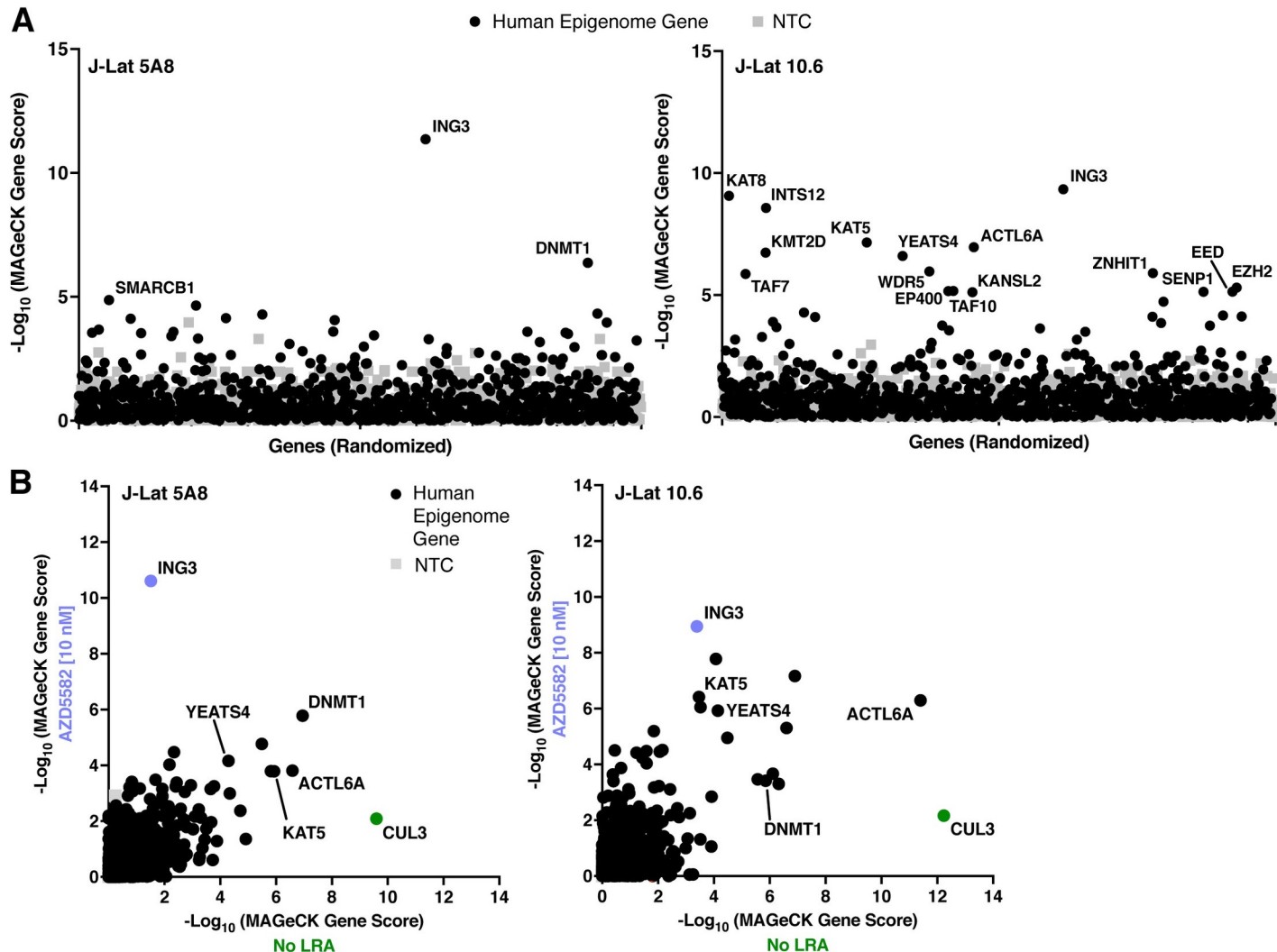

**Fig 5. LRA Latency HIV-CRISPR screen identifies *ING3* in combination with AZD5582 as a HIV-1 latency maintenance factor. (A)** *ING3* is the top hit of the LRA Latency HIV-CRISPR screen. The Latency HIV-CRISPR screen HuEpi knockout cells were treated with a low activating dose (10 nM) of AZD5582. The -log$_{10}$ of the MAGeCK score (on the y-axis) for each gene targeted by the HuEpi sgRNA library (black circles) and NTCs (gray squares) are calculated and displayed. Gene names are labelled for hits that have a <10% false discovery rate (FDR) in each J-Lat cell lines. NTCs are artificial NTC genes designed by iterative binning of NTC sgRNA sequences (see Methods). Genes are randomized on the x-axis, but the same order is used for both right and left panels. **(B)** Comparison of the Latency HIV-CRISPR screen by MAGeCK score in the presence (y-axis) and absence of AZD5582 (x-axis) with HuEpi genes (circles) and NTCs (gray squares). The data for the screen without an LRA (x-axis) is from Fig 3 as these two screens were performed in parallel. The genes unique to each screen are closest to the respective axis and the genes that are in common to both screens are at the center. *ING3* is highlighted in periwinkle and *CUL3* in green.

maintenance regardless of activation of the non-canonical NFκB pathway. We also observed that *CUL3*, the top hit unique to the screen in the absence of AZD5582, was relatively depleted in the screen conducted in the presence of AZD5582 (**Fig 5B**). This suggests a potential overlap between AZD5582 non-canonical NFκB function and *CUL3* function in the context of HIV-1 latency reversal. However, because *ING3* was the top gene hit unique to the screen in the presence of AZD5582, we sought to perform additional validation for this gene.

To validate *ING3*, the top hit enriched in the AZD5582 LRA Latency HIV-CRISPR screen, we generated *ING3* knockout cell lines in both J-Lat 10.6 and 5A8 cells. Simultaneously, cell lines transduced with the NTC sgRNAs were generated as a negative control. The pools of

knockout cells were treated with or without a low reactivating dose (10 nM) of AZD5582 for 24 hours and then viral reactivation was measured by HIV-1 reverse transcriptase activity. Based on screen results, we hypothesized that knockout of *ING3* alone would result in some level of viral reactivation compared to the transduction of NTC sgRNAs, but upon treatment with a low reactivating dose of AZD5582, viral reactivation would become even more pronounced. Consistent with our hypothesis, we observed that in both J-Lat cell lines, low reactivating dose AZD5582 treatment of the cell lines transduced with the NTC sgRNAs results in minimal viral reactivation, but treatment of the *ING3* knockout cell lines results in significant, increased viral reactivation (**Fig 6A**). These results suggest a mechanistic interplay between *ING3* inhibition and AZD5582 treatment during the reactivation of HIV-1 transcription. Moreover, we observed a similar trend in the primary CD4+ T cell HIV-1 latency model described previously. In CD4+ T cells from three healthy donors in which the latency model was established, we performed independent *AAVS1* and *ING3* knockouts, followed by 1 μM AZD5582 treatment, and observed significant viral reactivation in the combination condition of *ING3* knockout and AZD5582 treatment (**Fig 6B and 6C**). This enhancement of viral reactivation in two latency models suggests that there may be interplay between the function of *ING3* and the non-canonical NFκB pathway, which is targeted by AZD5582, both in cell lines and in primary CD4+ T cells.

As AZD5582 activates the non-canonical NFκB pathway, it is possible that *ING3* knockout could either enhance this activity or act through an independent pathway in the context of HIV-1 viral reactivation. To distinguish between these two possibilities, we measured the protein levels of NFκB2 products which includes p100, the cytoplasmic NFκB2 protein, and p52, the cleaved and active subunit that translocates to the nucleus. As expected, activation of the non-canonical NFκB pathway upon AZD5582 treatment results in an increase in the cleaved p52 product and a decrease in the p100 product in the control (NTC) cells (**Fig 6D**). We find that p52 levels are similar in the control and *ING3* knockout J-Lat cell lines (**Fig 6D**) suggesting that another pathway besides the non-canonical NFκB is the main driver resulting in HIV-1 reactivation. We also note that the p100 band in the J-Lat 5A8 *ING3* knockout cells is very faint and reduced compared to the J-Lat 5A8 control cells (**Fig 6D**). Altogether the LRA Latency HIV-CRISPR screen demonstrates the powerful potential for examining multiple mechanisms simultaneously to gain a more comprehensive understanding of the mechanisms underpinning HIV-1 latency and to further improve existing LRAs.

## The combination of *ING3* knockout and AZD5582 treatment uniquely results in significant increases of RNA-Pol2-S5p levels at the HIV-1 provirus

To better understand how AZD5582 treatment and *ING3* knockout act in combination to promote reactivation of latent HIV-1 proviruses, we examined the chromatin-related changes to the HIV-1 provirus by performing automated CUT&Tag [32]. Specifically, we performed genome-wide profiling of the J-Lat 10.6 cell line under four different conditions: (i) Transduction of NTC sgRNA with a treatment of DMSO (NTC + DMSO), the negative control (ii) Transduction of NTC sgRNA with a treatment of 10 nM AZD5582 (NTC + AZD5582) (iii) *ING3* knockout with a treatment of DMSO (ING3 KO + DMSO) (iv) *ING3* knockout with a treatment of 10 nM AZD5582 (ING3 KO + AZD5582).

ING3 is a known member of the NuA4 HAT complex, which functions to acetylate the histone H4 tail [52]. BRD4 interacts with acetylated H4 [9] and both isoforms of BRD4 are known to negatively regulate HIV-1 transcription [16, 17]. Therefore, our hypothesis is that significant viral reactivation identified only in the combination of *ING3* knockout and

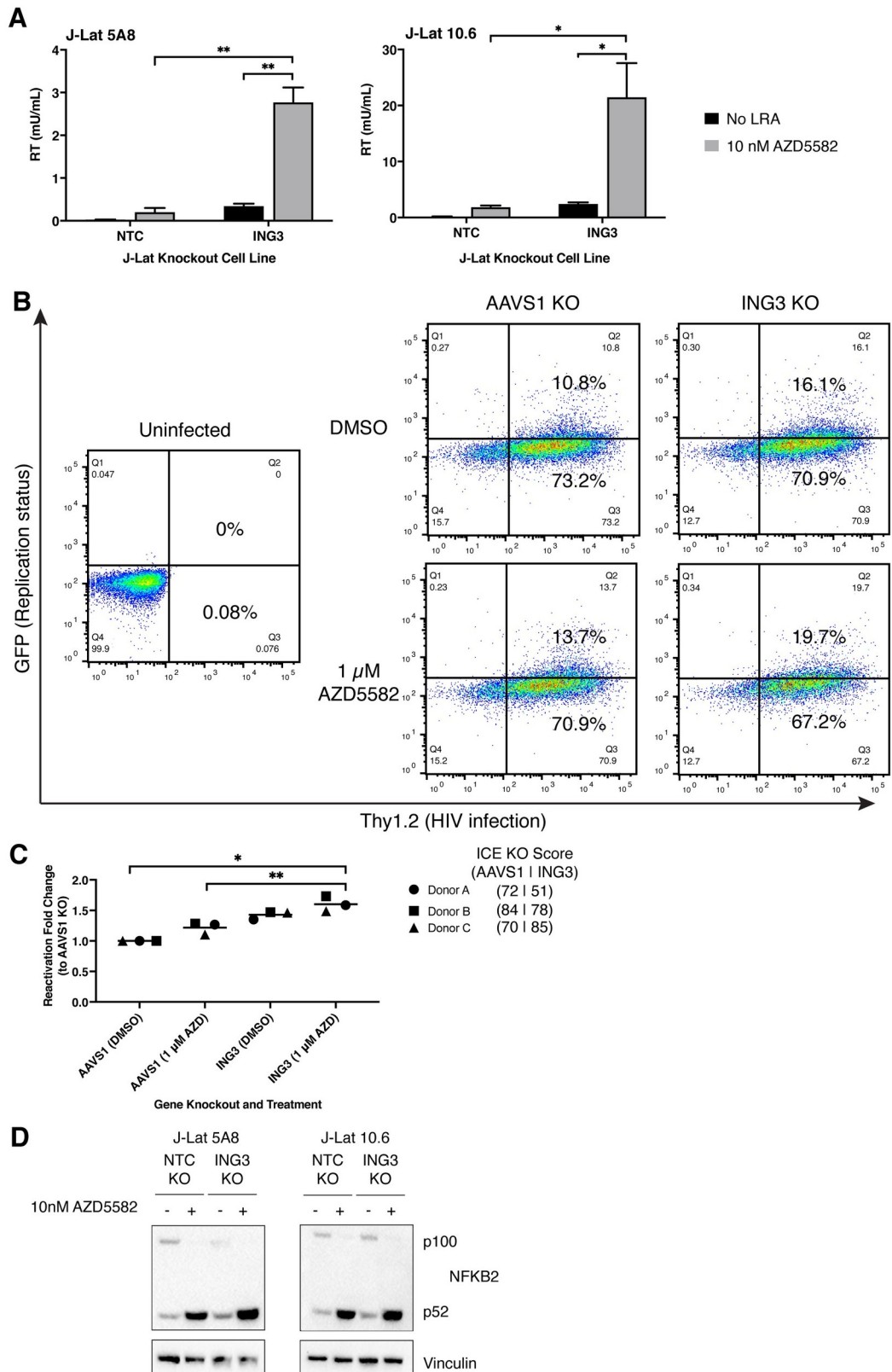

**Fig 6. Validation of ING3 in combination with AZD5582 as a HIV-1 latency maintenance factor. (A)** HIV-1 reverse transcriptase activity (y-axis) of the viral supernatant of NTC sgRNA transduction or *ING3* knockout in J-Lat 10.6 and 5A8

treated with 10 nM AZD5582 or an equivalent volume of DMSO. The ICE knockout score for the J-Lat 5A8 *ING3* knockout cell line is 72 and the ICE knockout score for the J-Lat 10.6 *ING3* knockout cell is 66. *ING3* knockout and AZD5582 treatment combine to result in a significant increase in viral reactivation. Paired t-test, p-value = <0.05 = *, = <0.01 = **. **(B)** Representative flow cytometry plots of primary CD4+ T cell HIV-1 latency model cells that are *AAVS1* or *ING3* knockouts treated with DMSO or 1 µM AZD5582 treatment. Thy1.2-, GFP- cells (quadrant 4) are uninfected; Thy1.2 +, GFP- (quadrant 3) cells are infected with the dual reporter HIV-1 virus and latent; Thy1.2+, GFP+ cells (quadrant 2) are infected and reactivated. **(C)** Independent knockouts of *AAVS1* and *ING3* in primary CD4+ T cell HIV-1 latency model cells were performed in three healthy donors and each pool of knockout cells were treated with DMSO or 1 µM AZD5582. Reactivation fold change is calculated based on percent Thy1.2+, GFP+ cells. The quantified percent Thy1.2+, GFP+ cells for each condition (knockout of *AAVS1* or *ING3* and treatment of DMSO or 1 µM AZD5582) was then normalized to the *AAVS1* knockout with DMSO treatment (negative control) condition. Knockout of *ING3* and AZD5582 treatment combined resulted in significant reactivation compared to knockout of *AAVS1*. Paired t-test, p-value = <0.05 = *. **(D)** Western blot showing similar p52 levels are detected in the control and ING3 knockout J-Lat 10.6 and 5A8 cell lines upon treatment of 10 nM AZD5582. Activation of the non-canonical NFκB (NFκB2) pathway is marked by a decrease in p100 and an increase in the cleaved product of p52.

AZD5582 treatment would be correlated with changes in levels of H4 acetylation, BRD4, and active RNA polymerase at the integrated provirus. As a result, we performed automated CUT&Tag using antibodies that recognized acetylated histone H4 (pan-H4Ac), BRD4, RNA-Pol2-S5p, RNA-Pol2-S2p, as well as a non-specific IgG negative control, at both the HIV-1 provirus genome and globally across the human genome.

We observed no change in pan-H4Ac signal over the HIV-1 LTR upon 10 nM AZD5582 treatment, but consistent with the known function of ING3, upon *ING3* knockout, there was a significant decrease in pan-H4Ac signal at the HIV-1 LTR (p = 0.0061) (**Fig 7A and 7B**). Additionally, the combination of *ING3* knockout with AZD5582 treatment, resulted in a significant decrease in pan-H4Ac signal at the HIV-1 LTR (p = 0.0415) (**Fig 7A and 7B**). Notably, in the *ING3* knockout condition alone, we observed a significant decrease in pan-H4Ac signal over both the U3 region of the HIV-1 LTR (p = 0.0070) as well as a significant decrease in the pan-H4Ac signal in the R and U5 region of the HIV-1 LTR (p = 0.0126) containing the HIV-1 provirus transcription start site (**Fig 7A and 7B**). When the *ING3* knockout is combined with AZD5582 treatment, there is only a significant decrease in the R and U5 region of the HIV-1 LTR (p = 0.0016) (**Fig 7A and 7B**). Thus, *ING3* knockout appears to be the main driver of the change in pan-H4Ac levels on the HIV-1 LTR in the presence and absence of AZD5582. We also find that that the reduction in pan-H4Ac levels we observed on the HIV-1 LTR in the *ING3* knockout condition correlate with reduced recruitment of BRD4. Indeed, as compared to NTC + DMSO, AZD5582 treatment alone did not affect BRD4 levels over the HIV-1 LTR; however, BRD4 levels were significantly decreased over the LTR both in the condition of *ING3* knockout alone and in combination with AZD5582 treatment (p = 0.0235, p = 0.0349, respectively) (**Fig 7C and 7D**). In the absence of AZD5582, the *ING3* knockout results in a significant decrease in the occupancy of BRD4 only over the U3 region of the LTR (p = 0.0219), while in the presence of AZD5582 combined with the *ING3* knockout, the decrease in BRD4 levels was only significant over the R and U5 region of the HIV-1 LTR (p = 0.0159) (**Fig 7C and 7D**). Altogether, our CUT&Tag results demonstrate that the pan-H4Ac and BRD4 levels are correlated at the HIV-1 LTR upon *ING3* knockout with and without AZD5582 treatment.

In contrast to the change in pan-H4Ac and BRD4 CUT&Tag levels at the HIV-1 LTR independent of LRA treatment, we find a striking difference in markers for transcription initiation and elongation as mediated by the phosphorylation of RNA Polymerase II C-terminal domain of Rpb1 (RNA-Pol2-S5p and RNA-Pol2-S2p, respectively) [53] only in the condition when the *ING3* knockout is combined with activation of the non-canonical NFκB pathway through AZD5582 treatment (**Fig 7E–7H**). Specifically, we find that there is a striking, significant increase in RNA-Pol2-S5p signal at the HIV-1 LTR (p = 4.2827E-5), body of the provirus

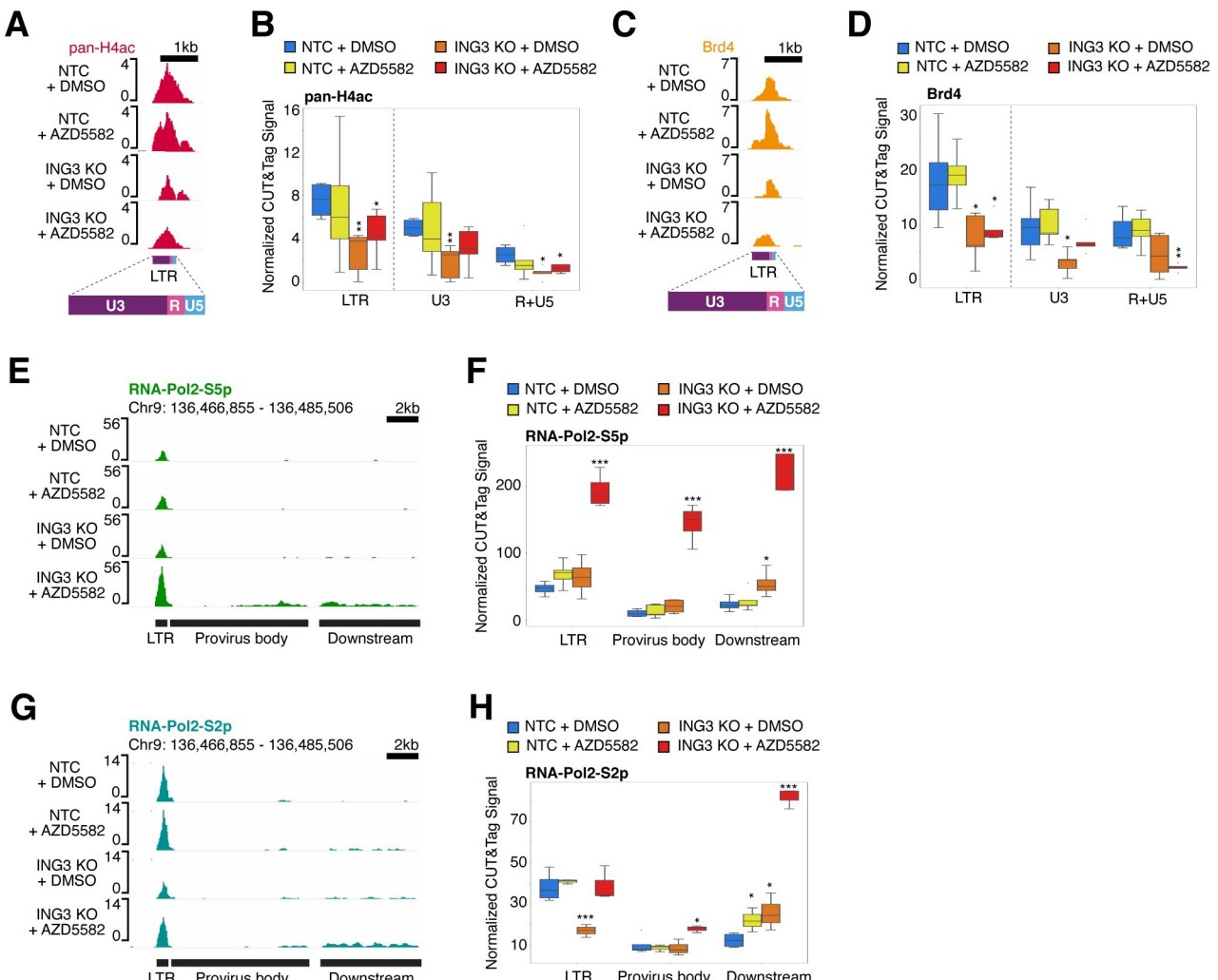

**Fig 7.** *ING3* **knockout decreases pan-H4Ac and BRD4 levels and stimulates HIV-1 transcriptional initiation and elongation upon addition of AZD5582. (A)** Genome browser tracks centered over the HIV-1 LTR showing the pan-H4Ac signal decreases in the *ING3* knockout alone and in combination with AZD5582 conditions. The y-axis represents read count. Because the HIV-1 LTR sequence is identical between 5' LTR vs. the 3' LTR, the CUT&Tag reads are combined onto one LTR. The LTR is subdivided into three regions: U3, R (containing the transcription start site), and U5. As quality control, we determined the signal levels of pan-H4Ac from the CUT&Tag data and confirmed the replicates of the antibody was most highly correlated amongst the pan-H4Ac replicates (**S3A Fig**). **(B)** Box plot showing the pan-H4Ac levels quantified over the full LTR including the U3 region and the R+U5 regions that include the transcriptional start site. The y-axis is the pan-H4Ac base pair coverage normalized to the total base pair coverage across the genome. Blue represents the transduction of NTC sgRNA with treatment of DMSO; yellow represents the transduction of NTC sgRNA with a treatment of 10 nM AZD5582; orange represents *ING3* knockout with treatment of DMSO; red represents *ING3* knockout with treatment of 10 nM AZD5582. Replicates for IgG n = 18 and pan-H4Ac n = 24. For statistical analysis, all conditions are compared to the NTC knockout and DMSO treatment control. P-value <0.05 = *, <0.005 = **. **(C)** Genome browser tracks showing BRD4 levels decrease in the *ING3* knockout alone and in combination with AZD5582 conditions. As quality control, we determined the signal levels of BRD4 from the CUT&Tag data and confirmed the replicates of the antibody was most highly correlated amongst the BRD4 replicates (**S3A Fig**). **(D)** Same as (B) but quantifying BRD4 levels. Replicates for BRD4 n = 16. **(E)** Genome browser tracks showing RNA-Pol2-S5p levels increase at the HIV-1 LTR, as well as the body of the provirus downstream of the 5' LTR ("provirus body"), and the region of the host genome downstream of the 3' LTR of the integrated provirus ("downstream") upon *ING3* knockout and AZD5582 treatment combined. As quality control, we determined the signal levels of RNA-Pol2-S5p from the CUT&Tag data and confirmed the replicates of the antibody was most highly correlated amongst the RNA-Pol2-S5p replicates (**S3B Fig**). **(F)** Box plot showing the quantification of the RNA-Pol2-S5p CUT&Tag signal over the HIV-1 LTR, the body of the provirus, and the region of the host genome downstream of the provirus. The y-axis is the RNA-Pol2-S5p base pair coverage normalized to the total base pair coverage across the genome. Blue represents the transduction of NTC sgRNA with treatment of DMSO; yellow represents the transduction of NTC sgRNA with a treatment of 10 nM AZD5582; orange represents *ING3* knockout with treatment of DMSO; red represents *ING3* knockout with treatment of 10 nM AZD5582. Replicates for RNA-Pol2-S5p n = 13. For statistical analysis, all conditions are compared to the NTC knockout and DMSO treatment control. P-value <0.05 = *, <0.005 = **, <0.0005 = ***. **(G)** Genome browser tracks showing RNA-Pol2-S2p levels increase over the body of the HIV-1 provirus as well as the host genome downstream of the provirus upon *ING3* knockout and AZD5582 treatment combined. As quality control, we determined the signal levels of RNA-Pol2-S2p from the CUT&Tag data and confirmed the replicates of the antibody was most highly correlated amongst the RNA-Pol2-S2p replicates (**S3B Fig**). **(H)** Same as (F) but quantifying RNA-Pol2-S2p levels. Replicates for RNA-Pol2-S2p n = 17.

("provirus body") (p = 8.943E-5), and the region downstream of the integrated HIV-1 provirus ("downstream") (p = 2.03E-5) only in the condition where cells that are knocked out for *ING3* are additionally treated with AZD5582 (**Fig 7E and 7F**). In contrast, for the RNA-Pol2-S2p signal, in cells with a combination of *ING3* knockout and AZD5582 treatment, we observed a significant increase in the RNA-Pol2-S2p signal in the provirus body (p = 0.0458) and downstream region (p = 3.89E-5), but not over the LTR (**Fig 7G and 7H**). Interestingly, in the intermediate condition of only *ING3* knockout, the signal of RNA-Pol2-S2p is significantly reduced at the HIV-1 LTR (p = 0.00583) (**Fig 7G and 7H**). This reduction in RNA-Pol2-S2p signal might be partially explained by the concomitant reduction we see in pan-H4Ac signal at the HIV-1 LTR in the *ING3* knockout condition alone. The striking increase in HIV-1 reverse transcriptase activity in the combination condition is concordant with our CUT&Tag results and suggest that the knockout of *ING3* combines with stimulation of the non-canonical NFκB pathway by AZD5582 to promote a potent increase in the RNA-Pol2-S5p and RNA-Pol2-S2p on the HIV-1 proviral genome.

As we observed the most significant RNA-Pol2-S5p level changes in the combined *ING3* knockout and AZD5582 treatment condition, we next asked whether the increasing RNA-Pol2-S5p levels was specific to the HIV-1 provirus or if this was occurring genome-wide. We compared the RNA-Pol2-S5p signal in the combined *ING3* knockout cells treated with AZD5582 to the control (NTC + DMSO) condition. We found the RNA-Pol2-S5p peaks over the HIV-1 LTR, provirus body, and downstream region were amongst the peaks with the most significant increase in RNA-Pol2-S5p signal (**Fig 8A**). We then rank ordered all the RNA-Pol2-S5p peaks by Log$_2$-fold change and found that out of the 2,547 total RNA-Pol2-S5p peaks, the top three peaks arranged from highest to lowest fold change were the HIV-1 provirus body, the region downstream of the HIV-1 integration site, and the HIV-1 LTR (**Fig 8B**). The two RNA-Pol2-S5p peaks with a significant increase in signal and the next highest fold

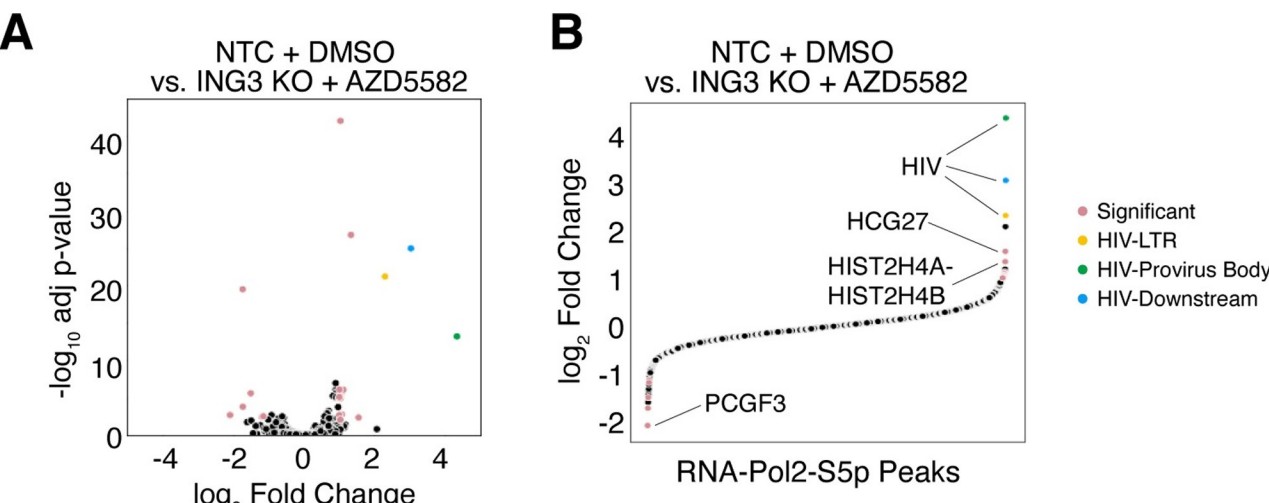

**Fig 8. Changes in RNA-Pol2-S5p levels upon *ING3* knockout combined with AZD5582 treatment are nearly unique to the HIV-1 provirus. (A)** Volcano plot comparing RNA-Pol2-S5p peaks between the NTC KO + DMSO (negative control) and ING3 KO + AZD5582 conditions. Highlighted peaks over the HIV-1 LTR (yellow circle), the body of the provirus (green circle), and downstream region of HIV-1 provirus (blue circle) are the regions of the highest RNA-Pol2-S5p fold change and are highly significant. Other regions with an absolute log$_2$ fold change greater than 1 and a -log$_{10}$ adjusted p-value > 2 are highlighted in pink. **(B)** Scatter plot showing all of the RNA-Pol2-S5p peaks rank ordered by fold change. Peaks are colored as in (A). The top three regions are the HIV-1 LTR, body of the HIV-1 provirus, and downstream region of the HIV-1 provirus. The next two regions overlap *HCG27* and the histone cluster 2 spanning HIST2H4A-HIST2H4B. The region with the greatest reduction in RNA-Pol2-S5p signal overlaps *PCGF3*.

change overlap HCG27, a long non-coding RNA, and HIST2H4A –HIST2H4B, part of histone cluster 2 in the human genome. The RNA-Pol2-S5p peak with the strongest reduction in RNA-Pol2-S5p signal in *ING3* knockout cells treated with AZD5582 overlaps PCGF3, a part of the Polycomb group PRC1-like complex (**Fig 8B**). Overall, this suggests the increase in transcription of the HIV-1 provirus caused by *ING3* knockout in combination with AZD5582 treatment, as correlated to the RNA-Pol2-S5p mark, is highly specific and nearly unique in the human genome. Thus, the CUT&Tag data in J-Lat 10.6 cells reveals that *ING3* knockout results in a reduction in H4Ac and BRD4 levels over the HIV-1 LTR and that *ING3* knockout acts in combination with AZD5582 treatment to dramatically increase the transcriptional initiation and transcription elongation of the HIV-1 provirus leading to latency reversal.

## Discussion

The hypothesis underlying this work is that modulating multiple overlapping mechanisms that control HIV-1 LTR-driven transcription will increase the potency and specificity of HIV-1 latency reactivation. That is, the goal is to find pathways that target potent transcriptional activation of latent HIV-1 proviruses with minimal global effects. We designed and validated a modular CRISPR screening approach which uses the incorporation of HIV-CRISPR genomes encoding sgRNAs into budding virions as a direct readout of activation of the latent provirus. The Latency HIV-CRISPR screen uses a sgRNA library of epigenetic regulatory genes and is paired with and without an LRA, AZD5582, to identify HIV-1 latency factors that act independently and in combination with AZD5582. We identified *ING3* as a gene whose knockout resulted in an enhanced increase of viral reactivation only in the presence of AZD5582. Using automated CUT&Tag, we observed that this enhancement is associated with active transcription marks as demonstrated by increased levels of RNA-Pol2-Ser5p and RNA-Pol2-Ser2p at the HIV-1 genome and mechanistically may be directly or indirectly dependent on a reduction in histone H4 acetylation and BRD4 occupancy over the HIV-1 LTR. We also validated CUL3 and multiple members of the NuA4 HAT complex as HIV-1 latency maintenance factors.

### Combinations of pathways with specificity for activating HIV-1 transcription

We find that two independent mechanisms—*ING3* knockout and treatment of a low-activating dose of AZD5582 –which when combined, have unique, specific effects on the HIV-1 LTR. These mechanisms function together to effectively activate HIV-1 transcription by increasing the presence of initiated and elongating RNA-Pol2 at the LTR and within the body of the HIV-1 provirus (**Fig 7**). Under these conditions used in our study, the RNA-Pol2-S5p peaks with the highest fold change throughout the entire genome were at the HIV-1 LTR and the provirus body (**Fig 8**). By CUT&Tag, RNA-Pol2-S5p appears as the primary marker for HIV-1 transcription in the J-Lat 10.6 cell line and this may be a result of P-TEFb, the major elongation factor for HIV-1 transcription, being a CTD kinase that can phosphorylate Ser5 and Ser2 individually, but not simultaneously [54]. Our model suggests that a reduction in NuA4 acetylation and BRD4-dependent occupancy at the HIV-1 LTR is associated with enhanced transcription in response to *ING3* knockout and activation of the non-canonical NFκB pathway.

We observed a reduction in histone acetylation and BRD4 recruitment to the HIV-1 LTR that is concomitant with enhanced RNA-Pol2 initiation and elongation only upon a combination of *ING3* knockout and AZD5582 treatment. This suggests the HIV-1 provirus may be subject to unexpected transcriptional control. In most mammalian transcriptional units driven by RNA-Pol2, NuA4 HAT is predominantly localized to promoters of active genes, specifically around the transcription start sites [55–57]. The NuA4 HAT preferentially acetylates histone

H4 [58] and this activity often impacts the regulation of gene expression by activating transcription. Additionally, BRD4 binds to acetylated chromatin [59], remains chromatin bound during mitosis [60, 61], and positively regulates mammalian transcription elongation through interaction with P-TEFb [37, 62]. Previous work has demonstrated that recruitment of other HAT complexes such as p300/CBP and P/CAF to the HIV-1 LTR followed by subsequent acetylation is associated with stimulation, rather than silencing, of HIV-1 transcription [63–69]. Moreover, using *in vitro* nucleosome-assembled templates in a Tat-independent system, the NuA4 HAT activates HIV-1 transcription [70]. Additionally, in the presence of Tat, the NuA4 HAT complex has been implicated in silencing of HIV-1 transcription. For example, in both a primary CD4+ T cell model and cells from ART-treated people living with HIV-1, the absence of KAT5, the catalytic component of the NuA4 HAT complex, promotes activation of HIV-1 transcription as a result of reduced H4 acetylation and BRD4 long isoform occupancy, and increased levels of the Super Elongation Complex (SEC) members at the HIV-1 LTR [9]. Furthermore, BRD4 has been demonstrated to not be required for HIV-1 transcription in the presence of Tat in HeLa cells [37] and has the potential to compete with Tat for binding to P-TEFb [16, 37]. Perhaps a competition model could explain why the reduced levels of acetylation and BRD4 we observed upon *ING3* knockout and AZD5582 treatment are associated with increases in HIV-1 transcription. Importantly, loss of BRD4 is not correlated with any changes in the binding of CDK9, a subunit of P-TEFb [71], thus the reduced levels of BRD4 are unlikely to be directly impacting P-TEFb binding. Instead, we propose a model in which the knockout of *ING3* leads to a decrease of H4 acetylation and BRD4, which enhances Tat recruitment of P-TEFb to TAR due to an increase in spatial opportunity for chromatin binding. In addition, the treatment of a low-activating dose of AZD5582 leads to recruitment of poised and active RNA-Pol2 at the HIV-1 LTR. Thus, upon the unique combination of *ING3* knockout and AZD5582 treatment, we observe the potent and nearly specific increase in viral transcription.

We observed a reduction in H4 acetylation across the entire HIV-1 LTR in response to *ING3* knockout, but in the presence of AZD5582 we see the most significant reduction of H4Ac signal and BRD4 occupancy at the R and U5 region which includes the HIV-1 transcription start site. This observation suggests, the increase in accessibility of the HIV-1 transcription start site when *ING3* knockout combined with AZD5582 treatment likely promotes both transcription initiation and elongation, culminating in the enhanced viral reactivation we observe in both J-Lat cells and primary CD4+ T cell HIV-1 latency models. Additionally, the dosage of AZD5582 used was a low-reactivating dose, suggesting that by inhibiting the NuA4 complex in parallel, it may be possible to efficiently activate HIV-1 transcription while causing minimal disruptions to a host system. HIV-1 is subject to complex transcriptional regulation, and these results highlight the importance of simultaneously examining multiple mechanisms to identify opportunities to specifically stimulate transcription of latent HIV-1 proviruses. These results are similar in concept to a recent study that found that the combination of AZD5582 and I-BET151 uniquely activated HIV-1 transcription in a Jurkat model of latency [72].

In support of the hypothesis that these mechanisms are nearly unique to the HIV-1 LTR, relatively few loci in the host genome showed a significant increase in RNA-Pol2-S5p occupancy when *ING3* knockout is combined with a low-activating dose treatment of AZD5582. One of these regions is downstream of the HIV-1 provirus integration site in J-Lat 10.6 cells and part of the *SEC16A* gene (**Fig 8**). Activation of human genes near or at the integrated provirus due to active transcription from an integrated provirus is a likely scenario as HIV-1 preferentially integrates into highly transcribed genes [73], and would be an unavoidable consequence of any reactivation strategy used for eradicating the latent HIV-1 reservoir. In the combined condition of *ING3* knockout and AZD5582 treatment, two other loci, a region of histone cluster 2 spanning HIST2H4A –HIST2H4B and a region overlapping with the long-

noncoding RNA, HCG27, also showed statistically significant, large fold changes in RNA-Pol2-S5p levels, but these changes were not as prominent as those observed over the HIV-1 provirus (**Fig 8**). In addition, we observed the gene PCGF3 had significantly reduced RNA-Pol2-S5p occupancy which could potentially to contribute to latency reversal as it is part of the non-canonical Polycomb group RING finger 3/5-PRC1 complex with silencing function [74] and is involved in the recruitment of PRC1 and PRC2 which have been shown to contribute to HIV-1 proviral silencing [8, 10, 75].

## Additional HIV-1 latency maintenance factors from the Latency HIV-CRISPR screen in the absence of LRAs

The top hit from our Latency HIV-CRISPR screen in the absence of LRA treatment was CUL3 which we also validated in a primary HIV-1 latency CD4+ T cell model. Our data is consistent with previous work as CUL3 has been demonstrated to negatively regulate HIV-1 transcription during productive infection directly or indirectly through the NFκB pathway as NFκB/ NFAT binding sites at the LTR are critical for regulating viral transcription [76]. Additionally, when we compared gene hits from Latency HIV-CRISPR screens performed in the presence and absence of AZD5582, we observed that CUL3 was preferentially decreased as a gene hit in the presence of AZD5582, which suggests pathway redundancy. Alternatively, ubiquitination has been shown to be involved in many aspects of HIV-1 transcriptional elongation such as ubiquitination of HEXIM-1 to release pTEFb from 7SK [77], ubiquitination of HIV-1 Tat protein by PJA2 to activate transcription [78], ubiquitination of factors (HCF1/2) to stabilize ELL2, a subunit of the Super Elongation Complex that stimulates transcriptional elongation [79]. CUL3 could play a novel role in the ubiquitination of one or more of these substrates.

In addition to CUL3, several other HIV-1 latency maintenance genes were also validated using two J-Lat models including ACTL6A and SRCAP (**Fig 4A**). *ACTL6A* knockout in both J-Lat 5A8 and 10.6 cells produced the highest levels of viral reactivation as measured by HIV-1 reverse transcriptase activity (**Fig 4A**). ACTL6A, also known as BAF53A, is a subunit of the NuA4 HAT complex, which our screen suggests altogether regulates HIV-1 transcription negatively (**Fig 3A**). ACTL6A is also a subunit of the BAF complex that has been shown to reposition nucleosome-1 resulting in lower DNA accessibility of the HIV-1 LTR which likely contributes to maintenance of a repressive state [80]. Additionally, ACTL6A is a subunit of the SRCAP complex that functions to exchange the histone H2A with histone variant H2A.Z [81] and the NuA4 HAT complex may enhance SCRAP activity [82]. Interestingly, our screen also identified hits unique to the SRCAP complex, like SRCAP (**Fig 3D**). We hypothesize that the role of ACTL6A in two relevant complexes likely accounts for the high level of HIV-1 reactivation in response to *ACTL6A* knockout. These results suggest that interplay of the SRCAP complex and histone modification by acetylation is critical for HIV-1 latency.

## Differences in HIV-1 latency models

The Latency HIV-CRISPR screen was established in the *in vitro* J-Lat cell line model as it is genetically tractable and a powerful tool to use to conduct genetic screens. However, as these cell lines are clonal in their virus integration sites and derived from leukemic cells with altered transcriptional and activation profiles, the cellular environment differs from that of primary CD4+ T cell models of HIV-1 latency. As a result, this could explain the discrepancy in results when comparing DMSO and AZD5582 treatment in *ING3* knockout cells which constitutes significant viral reactivation in J-Lat cells (**Fig 6A**), but not in our primary CD4+ T cell model of HIV-1 latency (**Fig 6B and 6C**). Also, the magnitude of viral reactivation in the combination of AZD5582 and *ING3* knockout levels in the primary CD4+ T cells (**Fig 6B and 6C**) is not as

striking as it is in both J-Lat cells (**Fig 6A**). However, the measure of viral reactivation in each model is different since the J-Lat cell model includes a nearly full-length provirus, which allowed us to measure viral reactivation based on HIV-1 reverse transcriptase activity of the virus particles produced in the viral supernatant. In contrast, this primary cell model uses a modified, reporter viral vector with premature stop codons in most HIV-1 genes to increase the efficiency of HIV-1 latency establishment [49], thus the measure of viral reactivation is based on intracellular reporter expression. As a result, the relevance of our results to *in vivo* HIV-1 latency reactivation remains to be tested but is nonetheless an entry into more potent and specific pathways towards "shock and kill" strategies. Overall, each model has differing strengths and as such we do not expect a full recapitulation of the trends observed.

## Comparison to other HIV latency screens

Our Latency HIV-CRISPR screening strategy is a unique approach to identify factors involved in HIV-1 latency. However, previous screens have also explored HIV-1 latency factors by screening using gene knockout, knockdown, or silencing [10, 21–28]. Generally, these previous screens used whole genome sgRNA libraries which make them unbiased for specific pathways, but this also limits the strength in statistical power to determine gene hits. Nonetheless, these previous screens have uncovered an exciting breadth of HIV-1 latency factors from the mTOR pathway [21] to proteasome related genes [24, 26] to the non-canonical NFκB pathway [28]. As epigenetics plays an important role in HIV-1 latency and it is still under investigation, we focused our Latency HIV-CRISPR screen for epigenetic regulatory factors by using a small, custom designed sgRNA library. Additionally, a smaller library can increase the dynamic range of a screen [83]. Our screen also relies on the HIV-CRISPR vector [31] to perform a high-throughput screen, which uses HIV-1 replication, rather than another reporter, as the direct readout of the screen. The difference in readout from other screens including screens with epigenetic hits [10, 23, 27] may result in different gene hits being highlighted. Another challenge of studying mechanisms of HIV-1 latency is the interaction of complex mechanisms in parallel and to address this using a CRISPR screen approach, we combined the screen with a low-activating dose of LRA treatment. By focusing our sgRNA library on epigenetic regulators and performing our screen with and without AZD5582 treatment, we identified a unique combination of modifications that synergize to stimulate HIV-1 transcriptional initiation and elongation.

A paper published while this manuscript was being prepared used a similar strategy of CRISPR screening in the presence of low-activating doses of LRA, but with a whole-genome guideRNA library [84]. Dai et al found that *BRD2* knockout synergized with AZD5582 [84], which likely falls within the same pathway identified in our study. In fact, in our screen results, *BRD2* also was a highly ranked gene hit in the presence of AZD6682 (rank #19 in the J-Lat 5A8 cell screen and #55 in the J-Lat 10.6 cell screen) (**S6 Table**). Thus, by exploring the combination of low-activating doses of LRAs with CRISPR based screens, the field can identify pathways leading to improvements in the specificity and potency of novel LRAs and advance towards a curative treatment strategy that eradicates the HIV-1 latent pool.

## Materials and methods

### Ethics statement

All primary cell data is from anonymous blood donors and is classified as "human subjects exempt" research by the Fred Hutchinson Cancer Center Institutional Review Boards, according to National Institutes of Health (NIH) guidelines (http://grants.nih.gov/grants/policy/hs/faqs_aps_definitions.htm). No animal work was done.

## J-Lat Wildtype, Clonal Knockout, and Pooled Knockout Cells

The HIV-1 latency cell line J-Lat 5A8 [30] and J-Lat 10.6 [29] were grown in RPMI-1640 medium (Thermo Fisher Scientific) supplemented with 10% Fetal Bovine Serum (FBS), Penicillin-Streptomycin (Pen/Strep), and 10 mM HEPES. In order to use these cells for screening with HIV-CRISPR, we performed CRISPR/Cas9-mediated knockout of Zinc Antiviral Protein (ZAP) [31] by electroporation of the J-Lat 10.6 and 5A8 cells with a Gene Knockout v2 kit (GKOv2) for ZAP (Synthego, Redwood City, CA) complexed with 1 μL of 20 μM Cas9-NLS (UC Berkeley Macro Lab). 5 days post electroporation, the cells were single cell sorted into a 96-well U-bottom plate (Sony MA900 Multi-Application Cell Sorter–Fred Hutch Flow Cytometry shared resource) and individual clones with biallelic knockouts of ZAP were used for subsequent Latency HIV-CRISPR screens. J-Lat cells with HIV-CRISPR KO pools targeting NFKBIA, KAT5, CUL3, ACTL6A, VPS72, DNMT1, DMAP1, SRCAP, YEATS4, and two non-targeting controls (S7 Table) were generated by transduction of lentivirus and subsequent 0.4 μg/mL puromycin selection for 10–14 days. To generate the CRISPR/Cas9-edited knockout pools, the sgRNAs that were the most overrepresented in the viral supernatant (highest combined p-value) of the Latency HIV-CRISPR screen were selected. Cell lines were determined to be mycoplasma free by the Fred Hutch Specimen Processing/Research Cell Bank shared resource.

## Plasmids

HIV-CRISPR plasmid was previously described [31]. HIV-CRISPR constructs targeting genes of interest were cloned by annealing complementary oligos (S7 Table) with overhangs that allow directional cloning into HIV-CRISPR using the BsmBI restriction sites. pMD2.G and psPAX2 plasmids were gifts from Didier Trono (Addgene #12259 and #12260, respectively). pMD2.Cocal plasmid was a gift from Hans-Peter Kiem [85]. lentiCRISPRv2 plasmid was a gift from Feng Zhang (Addgene #52961). The pNL4-3-Δ6-dreGFP-CD90 was previously described [50].

## Human Epigenome CRISPR/Cas9 sgRNA Library construction

The Human Epigenome (HuEpi) sgRNA library is composed of 841 genes of which 778 genes derive from the database Epifactor [33] and 63 genes were hand selected. For most genes, six sgRNA sequences were generated using GUIDES (Graphical User Interface for DNA Editing Screens) [86]. Two genes (HEATR1 and MCM2) have fourteen guides and one gene (UBE2N) has one guide. 252 non-targeting sgRNAs sourced from GeCKO v2.0 library [34] were also included resulting in a generated library of a total of 5,309 guides. The HuEpi sgRNA was synthesized (Twist Biosciences, San Francisco, CA) and cloned into HIV-CRISPR. Oligo pools were amplified using Phusion High-Fidelity DNA Polymerase (NEB) combined with 1 ng of pooled oligo template, primers ArrayF and ArrayR (ArrayF primer: TAACTTGAAAGTATT TCGATTTCTTGGCTTTATATATCTTGTGGAAAGGACGAAACACCG and ArrayR primer: ACTTTTTCAAGTTGATAACGGACTAGCCTTATTTTAACTTGCTATTTCT AGCTCTAAAAC), an annealing temperature of 59˚C, an extension time of 20 s, and 25 cycles. Following PCR amplification, a 140 bp amplicon was gel-purified and cloned into BsmBI (NEB; R0580) digested HIVCRISPR using Gibson Assembly (NEB; E2611S). Each Gibson reaction was carried out at 50˚C for 60 min. Drop dialysis was performed on each Gibson reaction according to the manufacturer's protocol using a Type-VS Millipore membrane (VSWP 02500). 5 μl of the reaction was used to transform 25 μl of Endura electrocompetent cells (Lucigen; 60242–2) according to the manufacturer's protocol using a Gene Pulser (BioRad). To ensure adequate representation, sufficient parallel transformations were

performed and plated onto carbenicillin containing LB agarose 245 mm x 245 mm plates (Thermo Fisher) at 300-times the total number of oligos of each library pool. After overnight growth at 37˚C, colonies were scraped off, pelleted, and used for plasmid DNA preps using the Endotoxin-Free Nucleobond Plasmid Midiprep kit (Takara Bio; 740422.10). The HuEpi library was sequenced and contains all 5,309 sgRNAs included in the synthesis (GSE215430).

## Lentivirus production

293T cells (ATCC;CRL-3216) cultured in DMEM (Thermo Fisher Scientific) supplemented with 10% FBS and PenStrep were plated at 2E5 cells/mL in 2 mL in 6-well plates one day prior to transfection. Transfection is performed using TransIT-LT1 reagent (Mirus Bio LLC; MIR2300) with 3 uL of transfection reagent per μg of DNA. For lentiviral preps, 293Ts were transfected with 667 ng lentiviral plasmid, 500 ng psPAX2, and 333 ng pMD2.G. One day post-transfection, media was replaced. Two- or three- days post-transfection, viral supernatants were filtered through a 0.2 μm filter (Thermo Scientific; 720–1320). For HuEpi library lentiviral preps, the same transfection of 293Ts was performed and supernatants from forty 6-well plates were combined and concentrated by ultracentrifugation. About 30 mL of supernatant are aliquoted into a polypropylene tube (Beckman Coulter; 326823) and underlaid with sterile-filtered 20% sucrose (20% sucrose, 1 mM EDTA, 20 mM HEPES, 100 mM NaCl, distilled water). Each of the polypropylene tubes are placed in a swinging bucket and spun in a SW 28 rotor at 23,000 rpm for 1 hour at 4˚C in a Beckman Coulter Optima L-90K Ultracentrifuge. Supernatants were decanted and pellets are resuspended in DMEM or RPMI over several hours at 4˚C. Concentrated lentivirus was used immediately or aliquots were made and stored at -80˚C. All lentiviral transductions were performed in the presence of 20 μg/mL DEAE-Dextran (Sigma-Alrdich; D9885). For generating pNL4-3-Δ6-dreGFP-CD90 stocks, the procedure was the same as the lentiviral preps except that 293Ts were transfected with 900 ng lentiviral plasmid, 450 ng psPAX2, and 150 ng pMD2.Cocal.

## LRA and no LRA Latency HIV-CRISPR Screening

The HuEpi library lentiviral preps were titered by a colony-forming assay in TZM-bl cells (NIH AIDS Reagent Program; ARP-8129) and used to transduce J-Lat 10.6 and J-Lat 5A8 cells at an MOI of 0.4. For the transduction, 3E6 cells per replicate (>500x coverage of the HuEpi library) was used and the spinoculation was performed at 1100xg for 30 min in the presence of 20 μg/mL DEAE-Dextran. Cells were selected in puromycin (0.4 μg/mL) for 10–14 days. In preparation for the LRA screen, AZD5582 dihydrochloride (Tocris; 5141) was resuspended in DMSO to 1 mM stocks. Subsequent dilutions are performed in RPMI and used immediately. Upon completion of selection, for the LRA screen, 3E6 cells per replicate per cell line were treated with 10 nM AZD5582 for 24 hours. Cells and supernatants were collected post selection or post treatment. Cells were washed once with DPBS (Gibco; 14190144) and cell pellets were stored at -20˚C. Genomic DNA was extracted from cell pellets with the QIAamp DNA Blood Midi Kit (Qiagen; 51183) and genomic DNA was eluted in distilled water. Viral supernatants were spun at 1100xg to remove cell debris, filtered through a 0.22 μm filter (Millipore Sigma, SE1M179M6), overlaid on a 20% sucrose cushion, and concentrated in a SW 28 rotor for 1 hour at 4˚C. The pellet is resuspended in RPMI and stored at -80˚C. Viral RNA was extracted from the concentrated virus with the QIAamp Viral RNA Mini Kit (Qiagen, 52904). The sgRNA sequences found in the genomic DNA and viral RNA samples were amplified by PCR (Agilent; 600677) and RT-PCR (Invitrogen; 18064014), respectively, using HIV-CRISPR specific primers. A second round of PCR is performed to barcode and prepare the libraries for Illumina sequencing (**S7 Table**). Each amplicon was cleaned up using double-sided bead

clean-up (Beckman Coulter; A63880), quantified with a Qubit dsDNA HS Assay Kit (Invitrogen; Q32854), and pooled to 10 nM for each library. Library pools are sequenced on a single lane of an Illumina HiSeq 2500 in Rapid Run mode (Fred Hutch Genomics and Bioinformatics shared resource).

### Screen analysis

Raw sequencing data is available as a GEO DataSet (GSE215430). Library pools are demultiplexed, reads are assigned to respective samples, trimmed, and aligned to the HuEpi library via Bowtie [87]. An artificial NTC sgRNA gene set the same size as the HuEpi library was generated by iteratively binning the NTC sgRNA sequences. Analysis of the screen to determine relative enrichment or depletion of the sgRNAs and genes were performed using the MAGeCK statistical package [36].

### Western blotting

Cells were harvested and washed once with ice cold phosphate-buffered saline (PBS) and pelleted by centrifugation (300 x g for 3 min). Whole cell lysate was extracted on ice by first resuspending the cell pellet with Pierce IP Lysis Buffer (ThermoFisher Scientific; 87787) supplemented with cOmplete protease inhibitor cocktail (Roche; 11697498001). The samples were then incubated on ice for 10 min with brief vortexing every 2–3 min and followed by a centrifugation at 16,000 x g for 20 min at 4°C. To prepare the samples, add 4x NuPAGE LDS Sample Buffer (ThermoFisher Scientific; NP0008) containing 5% 2-Mercaptoethanol (Millipore Sigma; M3148) and boil the samples at 95°C for 5 min. Lysates were resolved on a NuPAGE 4–12% Bis-Tris pre-cast gel (ThermoFisher Scientific; NP0336) and transferred to nitrocellulose membranes (Biorad; 1620115). Blocking was performed for 1 hr at room temperature using 5% milk/0.1% Tween-20 added to Tris-buffered saline (TBS) (20 mM Tris base and 150 mM NaCl at pH 7.6). Immunoblotting was performed using the primary antibodies CUL3 (Cell Signaling; 2759) at 1:1000, NFkB2 (Cell Signaling; 4882) at 1:1000, and viniculin (Santa Cruz; sc-25336) at 1:5000. Membranes were washed with TBST 6 times for 5 min each. The following secondary antibodies were used at a 1:5000 dilution: goat anti-rabbit IgG-HRP (R&D Systems; HAF008) and goat anti-mouse IgG-HRP (R&D Systems; HAF007). Membranes were developed with SuperSignal West Femto Maximum Sensitivity Substrate (Thermo Fisher; 34095) and visualized on a BioRad Chemidoc MP Imaging System.

### Genomic editing analysis

Knockout cells were harvested and DNA was extracted using the QIAamp DNA Blood Mini Kit (Qiagen; 51104). Primers from each targeted locus (**S7 Table**) were used to amplify the edited loci and the PCR was performed with either Platinum Taq DNA Polymerase High Fidelity (ThermoFisher Scientific; 11304011) or Q5 High-Fidelity DNA Polymerase (NEB; M0491S). Sanger sequencing was performed on PCR amplicons (Fred Hutch Genomics shared resource) using a sequencing primer (**S7 Table**) and results were analyzed by Inference of CRISPR Edits (ICE) [45] to determine gene editing outcome.

### Virus Release (RT assay)

Clarified viral supernatants are harvested and reverse transcriptase activity was measured using the HIV-1 reverse transcriptase (RT) activity assay as previously described [88, 89]. A standard curve is generated for all assays using a titered stock of HIV-1$_{LAI}$ that was aliquoted at and stored at -80°C.

## Primary CD4+ T Cell Latency Model and knockout

Used leukocyte filters from healthy donors were obtained from Bloodworks Northwest and total peripheral blood mononuclear cells (PBMCs) were isolated by purification over Ficoll (Millipore Sigma; GE17-1440-02). CD4+ T cells were isolated using magnetic negative selection (StemCell Technologies; 17952) and used immediately or frozen for storage in liquid nitrogen. CD4+ T cells were activated using anti-CD2, CD3, and CD28 beads (Miltenyi Biotec; 130-091-441) for 2 days and grown in RPMI-1640 medium (Thermo Fisher Scientific) supplemented with 10% Fetal Bovine Serum (FBS), Penicillin-Streptomycin (Pen/Strep), 1x Gluta-MAX (ThermoFisher Scientific; 35050061), 10 mM HEPES, 100 U/mL human interleukin-2 (Millipore Sigma; 11011456001), 2 ng/mL human interleukin-7 (Peptrotech; 200–07), and 2 ng/mL human interleukin-15 (Peprotech; 200–15). The beads were then removed by magnetic separation and cells were spinoculated with HIV-GFP-Thy1.2 (pNL4-3-Δ6-dreGFP-CD90) and 8 ug/mL polybrene (Millipore Sigma; TR-1003-G) for 2 hrs at 1100 x g. Spinoculated cells were then resuspended in fresh media with IL-2, IL-7, and IL-15 and incubated for 3 days before actively infected cells (Thy1.2+) were isolated by magnetic positive selection (Stem Cell Technologies; 18951). Infected cells (Thy1.2+) are maintained for 2 days before knockout of AAVS1, CUL3, and ING3 using GKOv2 kits from Synthego. crRNPs for each gene of interest were generated as described previously in the Knockout Cell Clones and Pools section except that supplemented P3 buffer (Lonza; V4SP-3096) is used instead of SE buffer. For each electroporation, 1.5E6 cells were pelleted by centrifugation at 100 x g for 10 min at 25˚C, washed once with PBS, pelleted again by centrifugation, PBS was removed, and resuspended with a crRNP complex. The resuspended cells were immediately transferred into the cuvette of the P3 Primary Cell Nucleofector Kit (Lonza; V4SP-3096) and electroporated using code EH-100 on the Lonza 4D-Nucleofector. 80 μL of prewarmed supplemented RPMI media with IL-2, IL-7, and IL-15 was added and cells were allowed to recover for 10 min in the 37˚C incubator. 300 μL of fresh media was added and cells are transferred to a 96-well plate. 200 μL of additional supplemented media was added 2 days later. The knockout cells are maintained at 1E6 cells/mL with fresh media supplemented with IL-2, IL-7, and IL-15. At 14 days post infection, cells are co-cultured with H80 feeder cells and maintained at 2E6 cells/mL in media supplemented with 20 U/mL IL-2. If AZD5582 treatment applies, cells are treated 1 uM AZD5582 media for 24 hours. At 19 days post infection, cells were stained with Thy1.2 antibody (Biolegend; 140323), fixed with 4% PFA, and underwent flow cytometry analysis (CD FACS Celesta–Fred Hutch Flow Cytometry shared resource).

## Automated CUT&Tag Profiling

We prepared nuclei from J-Lat 10.6 cells under four conditions: (i) NTC knockout with a treatment of DMSO (negative control) (ii) NTC knockout with a treatment of 10 nM AZD5582 (iii) ING3 knockout with a treatment of DMSO (iv) ING3 knockout with a treatment of 10 nM AZD5582. Up to 10 million cells were pelleted in 1.5 mL microfuge tubes spun at 300 x g for 10 min. Cells were then resuspended in 1 mL of ice cold NE1 Buffer (20 mM HEPES-KOH pH 7.9, 10 mM KCl, 0.5 mM Spermidine, 0.1% TritonX-100, 20% Glycerol, with Roche complete EDTA-free protease inhibitor tablet), and incubated on ice for 10 min. Nuclei were then centrifuged at 4˚C at 1,300 x g for 4 min. Nuclei were then resuspended in 1 mL of Wash Buffer (20 mM HEPES pH7.5, 150 mM NaCl, 0.5 mM spermidine, supplemented with Roche complete EDTA-free protease inhibitor tablet). 10 μL was used to determine the concentration of native J-Lat 10.6 nuclei, and nuclei were diluted to a concentration of 1 million nuclei/ 900 μL of Wash Buffer and 900 μL aliquots of this suspension was mixed with 100 μL of DMSO in Cryovials, which were then sealed and placed inside a Mr. Frosty isopropanol

chamber for slow freezing at -80˚C. Nuclei were then stored at -80˚C until use. For automated CUT&Tag processing, nuclei were thawed at room temperature, washed in wash buffer, and bound to concanvalin-A (ConA) paramagnetic beads (Bangs Laboratories; BP531) for magnetic separation as described on the protocols.io website (https://doi.org/10.17504/protocols.io.bgztjx6n). Samples were then suspended in antibody binding buffer and split for overnight incubation with antibodies specific to panH4Ac (Active Motif; 39925), BRD4 (Cell Signaling; 13440), RNA-Pol2-S5p (Cell Signaling; 13523), RNA-Pol2-S2p (Cell Signaling; 13499), and IgG control(Abcam; 172730). Sample processing was performed in a 96 well plate using 100K Con-A bound nuclei per reaction on a Beckman Coulter Biomek liquid handling robot according to the AutoCUT&Tag protocol available from the protocols.io website (https://doi.org/10.17504/protocols.io.bgztjx6n) and described previously [32] (Fred Hutch Genomics shared resource).

## CUT&Tag Sequencing Data Processing and analysis

For AutoCUT&Tag sample pooling and sequencing, the size distribution and molar concentration of libraries were determined using an Agilent 4200 TapeStation, and up to 96 barcoded CUT&Tag libraries were pooled at approximately equimolar concentration for sequencing. Paired-end 2 × 50 bp sequencing was performed on the NextSeq 2000 platform by the Fred Hutchinson Cancer Research Center Genomics Shared Resources. This yielded 5–10 million reads per antibody. Sequences the extended into the 3' adapter were first removed using the adapter clipping tool by cutadapt 2.9 with parameters:

-j 8—nextseq-trim 20 -m 20 -a AGATCGGAAGAGCACACGTCTGAACTCCAGTCA -A AGATCGGAAGAGCGTCGTGTAGGGAAAGAGTGT -Z.

To construct a reference reflecting integration of HIV-1 into J-Lat cells, we started with the UCSC hg38 human reference sequence from the Illumina iGenomes collection (https://support.illumina.com/sequencing/sequencing_software/igenome.html). Sequence for the integrated HIV-1 genome, including flanking human sequence, was obtained from accession MN989412.1 [90]. Accounting for flanking human sequence, we excised chr9:136468439– 136468594 from the reference Fasta file and inserted MN989412.1 in its place. We also rewrote gene & exon annotations downstream of the integration site, shifting all features in the associated GTF file by 10206bp to account for newly inserted HIV-1 sequence. Indexes for Bowtie2 v2.4.1 [91] and STAR v2.7.7a [92] were built using these modified Fasta and GTF files. The HIV-1 insertion is bounded by a pair of LTRs that, because they comprise identical sequence, are difficult for some alignment programs to interpret. To avoid these difficulties when needed, another version of the reference sequence was prepared with the second copy of the LTR sequence (634bp) masked by 'N's.

Fastq files were aligned to the custom hg38 genome with the HIV-1 genome inserted in chromosome 9 using Bowtie2 version 2.4.2 with the following parameters:—very-sensitive-local—soft-clipped-unmapped-tlen—dovetail—no-mixed—no-discordant -q—phred33 -I 10 -X 1000. Raw sequencing data is available as a GEO DataSet (GSE215430). Because the read depth and fraction of PCR duplicates varied considerably between replicates, we removed all duplicate reads from the Bed files. Peak calling was performed for the BRD4, pan-H4ac, RNA-Pol2-S5p, and RNA-Pol2-S2p data sets on the pooled reads from all replicates and all conditions using SEACR version 1.3 [93]. We called peaks using two settings, (1) in which using the IgG control data with stringent settings, and (2) using a FDR of 0.01, and then used the FDR 0.01 peaks that overlapped with IgG peaks as our final peak set. For correlation analysis between replicates, the BRD4 peak set was merged with the pan-H4ac peak set, and the RNA-Pol2-S5p peak set was merged with the RNA-Pol2-S2p peak set. For statistical

comparisons presented in Fig 7, we quantified the base pair coverage of pan-H4ac or BRD4 over the LTR, the U3 region, or the R + U5 region and these values were normalized to the total base pair coverage across the genome for each replicate. Similarly, for comparisons of the RNA-Pol2-S5p data and RNA-Pol2-S2p data, we quantified the base pair coverage over the LTR, Provirus Body, and Downstream Region of the HIV provirus these values were normalized to the total base pair coverage across the genome for each replicate. For comparisons in Fig 7, p-values were calculated using a two-sample t tests (two sided) with the SciPy.stats.ttest_ind() function in Python; P values were not corrected for multiple-hypothesis testing. For global comparison of RNA-Pol2-S5p data between the NTC + DMSO condition and the ING KO + AZD5582 condition, presented in Fig 8, reads that overlapped RNA-Pol2-S5p peaks were counted for each replicate and the log 2 Fold Change and adjusted p-values were calculated using DESeq2 version 1.32.0 using the Wald test. The adjusted p-values are corrected for multiple-hypothesis testing in a manner that is proportional to the number of RNA-Pol2-S5p peaks. DESeq2 assigns peaks with extremely sparse data are assigned an adjusted p-value of NaN, and these peaks were excluded form downstream analysis.

The numerical data used in all figures are included in **S8 Table** or available in GEO DataSet (GSE215430).

## Supporting information

**S1 Table. GuideRNA sequences for sgRNAs of the HuEpi library.**
(XLSX)

**S2 Table. MAGeCK analysis output by sgRNA for the Latency HIV-CRISPR screen.**
(XLSX)

**S3 Table. MAGeCK analysis output by gene or NTC for the Latency HIV-CRISPR screen.**
(XLSX)

**S4 Table. Literature analysis of top hit genes from the Latency HIV-CRISPR screen (listed in Fig 3B) to examine known functions of each gene in the context of HIV-1 latency. Includes supplemental references.**
(DOCX)

**S5 Table. MAGeCK analysis output by sgRNA for the AZD5582 LRA Latency HIV-CRISPR screen.**
(XLSX)

**S6 Table. MAGeCK analysis output by gene or NTC for the AZD5582 LRA Latency HIV-CRISPR screen.**
(XLSX)

**S7 Table. Primers sequences, sgRNA sequences, and next-generation sequencing barcodes.**
(XLSX)

**S8 Table. Spreadsheet containing the underlying numerical data for appropriate figure panels.**
(XLSX)

**S1 Fig. Validation of the top hit from the Latency HIV-CRISPR screen in primary CD4+ T cells. (A)** Representative flow cytometry plots of viral reactivation levels in wildtype primary CD4+ T cells and primary CD4+ T cell model of HIV-1 latency cells upon knockout of *AAVS1* and *CUL3*. Thy1.2-, GFP- cells (quadrant 4) are uninfected; Thy1.2+, GFP- (quadrant 3) cells are infected with the dual reporter HIV-1 virus and latent; Thy1.2+, GFP+ cells (quadrant 2)

are infected and reactivated.
(PDF)

**S2 Fig. AZD5582 dose curve in J-Lat cells. (A)** AZD5582 dose curve performed on both J-Lat 10.6 and 5A8 cell lines to determine viral reactivation levels.
(PDF)

**S3 Fig. The genome wide signal of automated CUT&Tag replicates is highly correlated. (A)** Correlation Matrix colored according to the pair-wise Pearson correlation of pan-H4Ac, BRD4, and IgG negative control samples across the merged pan-H4Ac and BRD4 peak sets. All pan-H4Ac samples group together by hierarchical clustering as do all of the BRD4 samples. **(B)** Same as (A) but showing the pair-wise Pearson correlation of RNA-Pol2-S5p and RNA-Pol2-S2p over the merged peak sets of these marks.
(PDF)

## Acknowledgments

We thank Carley Gray and Terry Hafer for critical feedback of this manuscript, Jessie Kulsuptrakul for technical assistance, and members of the Emerman lab for helpful suggestions. We thank Steve Hahn, Daphne Avgousti, and B. Matija Peterlin for helpful discussions. We thank Warner Greene at Gladstone Institute of Virology and Immunology and University of California, San Francisco, San Francisco, CA, USA for sharing the J-Lat 5A8 cells and Darell Bigner at Duke University, Durham, NC, USA for sharing the H80 cells. We thank Lucas Carter for assistance in assembly of the HuEpi library, Jackson Peterson for discussions on the primary cell HIV-1 latency model, Daryl Humes and Hannah Itell for discussions on primary cell protocols, and Jorja Henikoff, Matt Fitzgibbon and Pritha Chanana at the FHCC for bioinformatics support. Construction of the HuEpi library was supported by a Fred Hutchinson Cancer Center Pathogen Associated Malignancy Integrated Research Center award. This research was supported by the Genomics, Bioinformatics, and Flow Cytometry Shared Resource, RRID: SCR_022606, of the Fred Hutch/University of Washington Cancer Consortium (P30 CA015704).

## Author Contributions

**Conceptualization:** Emily Hsieh, Molly OhAinle, Michael Emerman.

**Data curation:** Emily Hsieh.

**Formal analysis:** Emily Hsieh.

**Funding acquisition:** Steve Henikoff, Michael Emerman.

**Investigation:** Emily Hsieh, Derek H. Janssens, Michael Emerman.

**Methodology:** Emily Hsieh, Derek H. Janssens, Patrick J. Paddison, Edward P. Browne, Steve Henikoff, Molly OhAinle.

**Project administration:** Michael Emerman.

**Resources:** Patrick J. Paddison, Edward P. Browne.

**Supervision:** Steve Henikoff, Molly OhAinle, Michael Emerman.

**Validation:** Emily Hsieh.

**Visualization:** Emily Hsieh.

**Writing – original draft:** Emily Hsieh, Derek H. Janssens, Michael Emerman.

**Writing – review & editing:** Emily Hsieh, Derek H. Janssens, Steve Henikoff, Molly OhAinle, Michael Emerman.

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
