## [Decision Letter · Decision Letter 0]

13 Dec 2022

Dear Dr. Emerman,

Thank you very much for submitting your manuscript "A modular CRISPR screen identifies individual and combination pathways contributing to HIV-1 latency" for consideration at PLOS Pathogens. As with all papers reviewed by the journal, your manuscript was reviewed by members of the editorial board and by several independent reviewers. The reviewers appreciated the attention to an important topic. Based on the reviews, we are likely to accept this manuscript for publication, providing that you modify the manuscript according to the review recommendations.

Sincerely,

Welkin E. Johnson

Academic Editor

PLOS Pathogens

Richard Koup

Section Editor

PLOS Pathogens

Kasturi Haldar

Editor-in-Chief

PLOS Pathogens

orcid.org/0000-0001-5065-158X

Michael Malim

Editor-in-Chief

PLOS Pathogens

orcid.org/0000-0002-7699-2064

Reviewer Comments (if any, and for reference):

Reviewer's Responses to Questions

**Part I - Summary**

Reviewer #1: Using a previously established HIV-CRISPR vector, Hsieh, E. et al. screened a custom library targeting genes known to regulate the epigenome for factors involved in the maintenance of HIV-1 latency and reactivation upon stimulation with a latency reversal reagent (LRA). This screen successfully identified several factors known to affect HIV-1 latency and a novel gene, ING3, knockout of which significantly increased HIV-1 expression specifically in the presence of the LRA AZD5582.

The screening approach employed is novel, and the analysis and follow-up on hits of interest were rigorously conducted. Overall, the data presented, and conclusions drawn were clear. Additionally, the unique effect of ING3 knockout/AZD5582 stimulation on HIV-1 transcription is both novel and interesting and demonstrates the value of this approach for probing for factors involved in latency. However, there are some areas in the text and figures that lack complete explanation or discussion of how the findings reported here fit into the existing literature on HIV-1 latency.

Reviewer #2: Hsieh et al., performed an elegant CRISPR-Cas library screen for latency reversal in two Jurkat cell clones. They focussed their screen on genes involved in epigenetic regulation in the presence and absence of AZD5582, a frequently used agent of latency reversal. Some of the major hits obtained in the absence of AZD5582 in both Jurkat cell clones had been reported previously thus functionally validating the screening approach. Major hits also belonged to the NuA4 and and SRCAP complexes suggesting a role for these complexes in latency. Analysis of single knock-outs were performed in the Jurkat cell clones to confirm the screening results. In the presence of AZD5582, they identified ING3 as the major hit and further explored latency reversal by AZD5582 and ING3 alone and in combination in a primary cell model of latency. The experiments are well controlled and presented in a logical and understandable manner.

**Part II – Major Issues: Key Experiments Required for Acceptance**

Reviewer #1: 1.The description of the screen design and conduct in the Results section does not specify that both J-Lat models used were actually clonal ZAP knock-outs. To the general reader this might be somewhat misleading, especially since later CRISPR knock-outs are specified to be pools of cells to “avoid clone-to-clone differences” (Lines 215-216). Although the generation of ZAP knock-out cells is described in the Materials and Methods, this is an important piece of information regarding the screen and the fact that modified J-Lats were used for these screens should be stated in the main text.

2.A clearer description of what has already been determined regarding the factors identified here would be helpful to put this investigation in context with the existing literature. For readers that are not experts in HIV-1 latency, it may be somewhat confusing what genes listed in Figure 3 have previously been identified versus those that were unique to this screen. Perhaps a supplemental table with references for the genes shown in Figure 3B would be appropriate. Did the authors find any novel factors in the screens without LRA addition? Are all the factors shown in Figure 4 previously implicated in HIV-1 latency?

3.The results concerning the role of ING3 in primary cells presented in Figure 6B-C do not coincide with the results from J-Lats, but the discrepancy is not discussed in the text. Specifically, there is no significant difference between DMSO and AZD treatment in the ING3 knock-out cells, as opposed to the large differences shown in Figure 6A. While the results in primary cells are not necessarily expected to fully recapitulate the J-Lat model, the authors should at a minimum note the discrepancy and ideally discuss potential explanations.

Reviewer #2: The relevance of the manuscript greatly depends how good the Jurkat cell model reflects latency reversal in primary cells. Therefore, the degree of latency reversal in primary CD4 T cells, as assessed in Fig. 6B, C should be reported and discussed more clearly. The representative flow cytometry plots should also contain a graph demonstrating the precentage of Thy1.2+ GFP+ cells before the cells have reached latency (e.g. data after the selection of Thy1.2 positive cells). It is also unclear, how precisely the „reactivation fold change“ in Fig. 6C was calculated. Is this based on percent GFP+ cells or does it include fluorescence intensities of the GFP+ cell population? It should be discussed why the fold-change reported in this assay, is not consistent with „dramatic increase“ stated in line 443 for the CUT + Tag data.

**Part III – Minor Issues: Editorial and Data Presentation Modifications**

Reviewer #1: 1.The justification and explanation for what genes were labeled in Figure 3A is excellent, but Figure 5A, which presents similar data, does not include this type of labeling. Although it is clear that the purpose of this figure is to identify factors specific to the screen conducted in the presence of the LRA, it would still be valuable to label genes that meet the same criteria as in 3A.

2.Lines 163-164 – How do the authors define what a “significant enrichment” is in reference to Figure 2A? Is there a fold-change cut-off? Is there a way to indicate this in the figure?

3.An interaction between CUL3 and NF-κB/NFAT transcription factor binding sites in the HIV-1 LTR has been previously shown (PMID:32882949), it would be worthwhile to briefly discuss how the findings presented here regarding CUL3 relate to this previous report for the overall understanding of how non-canonical NF-κB signaling affects HIV-1 transcription.

4.There are several portions of the text that are somewhat difficult to read due to awkward use of past/present tense. For example, Line 262 should read “AAVS1 gene was targeted”. A revision with careful attention to verb tenses would improve readability.

5.Line 175 – This is strange phrasing, control guides are non-targeting, not “targeting NTCs”.

6.Line 462 – Is it appropriate to say these factors were “identified” here? Perhaps “verified” or “validated” would be more accurate.

7.What is the ICE score for the ING3 knock-out cells shown in Figure 6A?

8.There appears to be no p100 band in the ING3 KO 5A8 cells in the absence of AZD treatment as opposed to the control cells. Is there reduced NFKB2 expression in these cells?

9.It is somewhat difficult to identify the colored dots in Figure 8. The authors might consider use of color-blind friendly schemes (such as the Paul Tol schemes) for figures that require the reader to discriminate between data points based on color to interpret points not specifically highlighted by the authors.

Reviewer #2: The introduction of latency should also include a reference to silencing mechanisms prior to integration.

Fig. 2B. Legend should state that data points represent individual guide RNAs targeting the same gene

Fig. 4 and 6A. The results of the RT-assay may depend on replication of the reactivated provirus. Therefore, it is essential to describe how many days after knock-out RT activity was detemined. The large differences in the degree of RT activation between the two cell clones also deserves a comment

PLOS authors have the option to publish the peer review history of their article (what does this mean?). If published, this will include your full peer review and any attached files.

Reviewer #1: No

Reviewer #2: No

Figure Files:

Data Requirements:

Reproducibility:

References:

---

## [Editor Report · Decision Letter 1]

5 Jan 2023

Dear Dr. Emerman,

We are pleased to inform you that your manuscript 'A modular CRISPR screen identifies individual and combination pathways contributing to HIV-1 latency' has been provisionally accepted for publication in PLOS Pathogens.

Best regards,

Welkin E. Johnson

Academic Editor

PLOS Pathogens

Richard Koup

Section Editor

PLOS Pathogens

Kasturi Haldar

Editor-in-Chief

PLOS Pathogens

orcid.org/0000-0001-5065-158X

Michael Malim

Editor-in-Chief

PLOS Pathogens

orcid.org/0000-0002-7699-2064
---

## [Editor Report · Acceptance letter]

24 Jan 2023

Dear Dr. Emerman,

We are delighted to inform you that your manuscript, "A modular CRISPR screen identifies individual and combination pathways contributing to HIV-1 latency," has been formally accepted for publication in PLOS Pathogens.

Best regards,

Kasturi Haldar

Editor-in-Chief

PLOS Pathogens

orcid.org/0000-0001-5065-158X

Michael Malim

Editor-in-Chief

PLOS Pathogens

orcid.org/0000-0002-7699-2064